# Score Correction for Generative Models with Probabilistic Constraints

**Shishang Wu** [1]  **Bingjing Tang** [2]  **Vinayak Rao** [1]

## Abstract

We introduce DUALSCORE, a framework for correcting score functions so that score-based generative models satisfy probabilistic constraints. These constraints are specified through the marginal distribution of a stochastic transformation of the modeled variable. We formulate this as a constrained KL-minimization problem, and optimize its dual, parameterizing the infinite-dimensional dual variable with a neural network. This yields an additive correction to the original score function that can be used directly for sampling via stochastic gradient Langevin dynamics or reverse diffusion sampling, without modifying the base model. We evaluate DUALSCORE on synthetic tasks and two real-world applications: regularized nonparametric maximum likelihood estimation and the incorporation of class-level constraints such as fairness into pretrained image diffusion models.

## 1. Introduction

Score-based models and algorithms have become central to learning and sampling from complex, high-dimensional distributions. Diffusion models (Song et al., 2021; Ho et al., 2020; Karras et al., 2022) learn the score of a data-generating distribution with a neural network, enabling state-of-the-art generation of images, audio and molecular structures. Score functions also underpin Langevin dynamics and stochastic gradient Langevin dynamics (SGLD, Welling & Teh (2011)), widely used for posterior simulation from Bayesian (including hierarchical and nonparametric) models (Gelman et al., 2013; Ghosal & Van der Vaart, 2017).

At a high level, the score function provides a representation of a probability distribution $p_X(x)$, one that is either learned

from data or that is specified by a statistical modeler. In modern applications, these distributions are often highly complex, and may encode behavior inconsistent with prior knowledge or domain-specific requirements. Such knowledge is often naturally formulated as a constraint on the distribution of a lower-dimensional variable $Y$, obtained from $X$ through a known, possibly stochastic, transformation $p_{Y|X}(y|x)$. For example, applying a classifier to generated images induces a marginal distribution over class labels. The practitioner may know or require that these induced marginals should match a target $p_Y^1$. In this work, we develop a framework for enforcing such distributional constraints through corrections to the score function.

We impose three desiderata on our framework:

D1. The corrected model of $X$ satisfies the distributional constraint on $Y$.

D2. The correction is minimal: the corrected model remains as close as possible to the original.

D3. No retraining of the original model is required.

We consider two motivating examples in this paper:

**P1: Fairness constraints in diffusion models.** An image diffusion model may inherit class imbalances from its training data. An attribute classifier $p_{Y|X}$ maps generated images to class labels, inducing a biased marginal $p_Y^0$ over classes. We seek to correct the diffusion model's score so the labels follow a prescribed distribution $p_Y^1$ over classes, without retraining the model. Class labels may themselves be ambiguous or latent, leading to imperfect classifiers.

**P2: Regularized nonparametric maximum likelihood estimation.** Given noisy observations $\{y_i\}$ of latent variables $\{x_i\}$ via some known measurement model $p_{Y|X}$, the NPMLE (Laird, 1978; Lindsay, 1995) seeks to nonparametrically recover the latent distribution over $X$. While flexible, the resulting estimate is often discrete, and does not incorporate prior knowledge. Wilkins-Reeves et al. (2021) regularize the estimate toward a reference distribution $p_X^0$. In our framework, this corresponds naturally to minimally modifying $p_X^0$ so that the marginal induced by $p_{Y|X}$ matches the observed distribution $p_Y^1$.

Similar constraint-correction problems arise in a wide range of applications, including simulation-based modeling, synthetic data generation, and privacy-constrained learning.

[1]Department of Statistics, Purdue University, West Lafayette, USA [2]Department of Epidemiology and Biostatistics, University of California, San Francisco, San Francisco, USA. Correspondence to: Shishang Wu <wu1396@purdue.edu>, Vinayak Rao <varao@purdue.edu>.

*Proceedings of the 43rd International Conference on Machine Learning*, Seoul, South Korea. PMLR 306, 2026. Copyright 2026 by the author(s).

**Contributions:** Our framework applies to settings where domain knowledge is expressed as a *distributional constraint* on a *stochastic transformation* of a score-based model. We refer to these together as *probabilistic constraints*. Prior work has mostly been restricted to deterministic transformations, which we show have qualitatively different behavior. Specific contributions are:

- We prove that importance reweighting and conditional sampling, the standard solutions for deterministic constraints, cannot in general satisfy the marginal constraint under stochastic $p_{Y|X}$. This motivates a new approach.

- We formulate our problem as a constrained KL-minimization, and leverage a dual formulation that yields an additive score correction. By parameterizing the dual variable with a lightweight network, we reduce a constrained infinite-dimensional problem to neural network training. No retraining of the original model is required.

- The learned score correction is used for post-training simulation via SGLD or reverse diffusion, making it directly usable as a plug-in for a range of score-based methods. This unifies fitting and sampling into a single framework.

- By grounding simulation in score-based dynamics, our method can remain effective when the reference and solution distributions have low overlap. This *density chasm* regime (Rhodes et al., 2020) is where importance-based approaches fail due to low effective sample size.

## 2. Problem Setup

Let $p_X^0(x)$ be the probability density of a random variable $X \in \mathcal{X}$ implied by some black-box generative model such as a diffusion model or some other prior distribution. We will call $p_X^0$ the *reference density*. Although our results apply more generally, our main algorithm requires access to the score $\nabla_x \log p_X^0(x)$ of the reference density. This requirement is satisfied when $p_X^0$ has an analytically tractable density, can be differentiated efficiently via automatic differentiation or corresponds to a learned score-based model.

Let $Y \in \mathcal{Y}$ be a random variable whose conditional distribution given $X$, $p_{Y|X}(y|x)$, is known. We refer to this as the *transformation density*. This can result from a known, lossy and potentially noisy transformation $Y = f(X, \xi)$ of $X$, where $\xi$ is some auxiliary random variable. Typically $Y$ is lower dimensional than $X$, and can even be categorical (in this case, we continue to use densities, now with respect to the uniform discrete measure).

Together, the reference density $p_X^0(x)$ and the transformation density $p_{Y|X}(y|x)$ induce a *marginal density* $p_Y^0(y) = \int_{\mathcal{X}} p_{Y|X}(y|x)p_X^0(x)dx := p_{Y|X} \circ p_X^0$ on $Y$. Often, the modeler has knowledge or has requirements about how $Y$ is distributed. We write this as a known density $p_Y^1(y)$, the

*constraint density*. Examples include:
a) $Y$ corresponds to a smaller component of some larger stochastic system $X$ whose behavior is well understood,
b) $Y$ is constrained by factors like fairness or privacy to follow some prescribed distribution,
c) $Y$ corresponds to noisy measurements of $X$, $p_Y^1$ is the (smoothed) empirical distribution of a finite dataset of $Y$'s.

Typically, the $p_Y^0$ implied by $p_X^0$ will not agree with $p_Y^1$, and we seek a distribution $p_X^*$ that is as close as possible to $p_X^0$ (in terms of KL divergence), while ensuring that the induced marginal $p_Y^* := p_{Y|X} \circ p_X^*$ is close to the constraint density $p_Y^1$. Formally, for some fixed $\varepsilon \geq 0$, we wish to solve

$$
\begin{aligned}
p_X^* = \underset{p_X \in \mathcal{P}_\mathcal{X}}{\arg\min} \quad & D_{\mathrm{KL}}(p_X \| p_X^0) \\
\text{subject to} \quad & \|p_Y^1 - p_{Y|X} \circ p_X\|_2 \leq \varepsilon.
\end{aligned}
\tag{1}
$$

Above $\mathcal{P}_\mathcal{X}$ is the space of all probability densities on $\mathcal{X}$. Setting $\varepsilon = 0$ seeks to enforce the constraint exactly, with larger values tolerating larger deviations.

The optimization problem above is not always feasible. Two simple counterexamples are: 1) if the union of the support of $p_{Y|X}(\cdot|x)$ across all $x \in \mathcal{X}$ is strictly contained within the support of $p_Y^1(y)$, and 2) if the variance introduced by $p_{Y|X}(y|x)$ exceeds the variance of $p_Y^1(y)$. More generally, for this problem to be feasible, $p_Y^1(\cdot)$ must lie in the convex hull of $p_{Y|X}(\cdot|x), x \in \mathcal{X}$, and with few exceptions, this is not easy to check. One exception is when $Y$ is a deterministic function of $X$; now, a sufficient condition for feasibility is that the range of the function $f$ equals the support of $p_Y^1$.

For any pair $(p_{Y|X}, p_Y^1)$, we define the associated *constraint gap* as $\varepsilon_* := \inf_{p_X \in \mathcal{P}_\mathcal{X}(p_X^0)} \|p_Y^1 - p_{Y|X} \circ p_X\|_2$, where $\mathcal{P}_\mathcal{X}(p_X^0) = \{p_X \in \mathcal{P}_X \text{ s.t. } D_{\mathrm{KL}}(p_X \| p_X^0) < \infty\}$. We say Problem (1) is *feasible* if $\varepsilon \geq \varepsilon_*$, and *infeasible* otherwise.

In real applications, $\varepsilon_*$ is unknown, and constraints are typically understood in a "the tighter, the better" manner. One can then try to find $\varepsilon_*$ via grid or annealing search. While we primarily focus on the feasible regime, our proposed dual optimization approach shows promising empirical performance even in the second regime.

## 3. Methodology

Directly solving equation 1 presents significant challenges: it is an infinite-dimensional optimization problem subject to two constraints: the marginal constraint on $p_Y$ and the requirement that the optimization variable lies on the infinite-dimensional probability simplex. Rather than attempting to solve it directly, we apply Lemma 11 of Altun & Smola (2006), adapted to our setting below, allowing us to write down and optimize a corresponding Fenchel dual problem:

**Lemma 3.1.** *Let $p_X^0$ and $p_Y^1$ be two probability densities on domain $\mathcal{X}$ and $\mathcal{Y}$, respectively, and $p_{Y|X} : \mathcal{X} \times \mathcal{Y} \to \mathbb{R}^+$*

*be a conditional probability density. For any $\varepsilon \geq 0$, we have*

$$\min_{p_X \in \mathcal{P}_{\mathcal{X}}} \left\{ D_{\mathrm{KL}}(p_X \| p_X^0) \text{ s.t. } \left\| p_Y^1 - p_{Y|X} \circ p_X \right\|_2 \leq \varepsilon \right\} \quad (2)$$

$$= \max_{\phi \in L^2(\mathcal{Y})} \left\{ \langle \phi(y), p_Y^1(y) \rangle - \varepsilon \|\phi\|_2 + e^{-1} \right.$$

$$\left. - \log \int_{\mathcal{X}} p_X^0(x) \exp \left( \langle \phi(y), p_{Y|X}(y|x) \rangle \right) \, dx \right\} \quad (3)$$

*When (2) is feasible (i.e., $\varepsilon \geq \varepsilon_*$), the unique solution is*

$$p_X^*(x) = p_X^0(x) \exp \left( \int_{\mathcal{Y}} \phi^*(y) p_{Y|X}(y|x) \, dy \right) \Big/ C_{\phi^*} \quad (4)$$

*where $\phi^*$ solves (3) and $C_{\phi^*}$ is the normalizing constant.*

Note that when (2) is infeasible, the dual is unbounded, and a dual optimization that diverges helps identify feasibility. Before discussing how we optimize the dual, we mention some implications of this result. For the special case of $\varepsilon = 0$, and a deterministic transformation $Y = f(X)$, so that $p_{Y|X}(y|x) = \delta_{y=f(x)}$, the above reduces to:

**Corollary 3.2.** *Define the $\varepsilon$ $p_Y^1$-approximation to $p_Y^0$ as $\widetilde{p}_Y^\varepsilon = \arg\min_{\|p_Y - p_Y^1\|_2 \leq \varepsilon} D_{\mathrm{KL}}(p_Y \| p_Y^0)$ (assume it exists). When $\varepsilon = 0$, $\widetilde{p}_Y^\varepsilon = p_Y^1$. Then, in the case of a deterministic transformation, where $p_{Y|X} = \delta_{y=f(x)}$ for some function $f : \mathcal{X} \to \mathcal{Y}$, the solution to Problem (1), is given by $p_X^*(x) = p_X^0(x) \frac{\widetilde{p}_Y^\varepsilon(f(x))}{p_Y^0(f(x))} = \widetilde{p}_Y^\varepsilon(f(x)) p_{X|Y}^0(x|f(x))$.*

For $\varepsilon = 0$, there are a few instances of this result being used either explicitly or implicitly (Kessler et al., 2015; Choi et al., 2020; Tang, 2023; Tang & Rao, 2025). We can exploit this to produce two simple approaches to solve Problem (1):

**Conditional sampling (CondSmpl):** first sample $Y$ from $p_Y^1$ (so the constraint is satisfied), and then, conditionally simulate $X$ from the posterior $p_X^0(x) p_{Y|X}(y|x) / p_Y^0(y)$. For instance, when $Y$ is a class label and $X$ an image, one might sample a class and then a class-conditioned image.

**Importance resampling (IRS):** estimate the marginal distributions over $Y$, and use these as importance weights to correct samples from the reference distribution $p_X^0$. Since $p_Y^1$ is given, we only need to estimate the marginal density $p_Y^0$ of the reference $p_X^0$.

Both these schemes can fail to satisfy the marginal constraint in the general setting with stochastic transformations. For example, let $p_X^0 = \mathcal{N}(0, 1^2)$, $p_{Y|X=x} = \mathcal{N}(2x + 1, 1^2)$ and $p_Y^1 = \mathcal{N}(-1, 2^2)$. The optimal solution for $\varepsilon = 0$ can be shown to be $p_X^* = \mathcal{N}(-1, \frac{3}{4})$ (see Appendix D.2.1). However, the conditional simulation approach yields $p_X^{\det}(x) = \mathcal{N}(-\frac{4}{5}, \frac{21}{25})$, resulting in $p_Y^{\det}(y) := \int_{\mathcal{X}} p_{Y|X}(y|x) \, p_X^{\det}(x) dx = \mathcal{N}(-\frac{3}{5}, \frac{109}{25}) \neq p_Y^1$. In general, we have the following result:

**Proposition 3.3.** *Define the Markov operator $\mathcal{A}$ : $L^2(p_Y^0) \to L^2(p_Y^0)$ as $(\mathcal{A}f)(y) := \mathbb{E}[\mathbb{E}[f(Y'|X)]|Y =$*

$y] = \int \left( \int p_{Y|X}(y'|x) f(y') \, dy' \right) \frac{p_{Y|X}(y|x) p_X^0(x)}{p_Y^0(y)} \, dx$. *Then:*
*a) $\mathcal{A}$ has eigenvalues in $[0, 1]$ with largest eigenvalue 1.*
*b) Let $\lambda_1$ denote its second largest eigenvalue, so that $1 - \lambda_1$ is the spectral gap. Let $\pi_{\min} \leq p_Y^0(y) \leq \pi_{\max}$ $\forall y \in \mathrm{supp}(p_Y^0)$. Then we can bound the induced marginal $p_Y^{det}$ of the deterministic scheme away from the target $p_Y^1$ as*
$\|p_Y^1 - p_Y^{det}\|_2 \geq \frac{\sqrt{\pi_{\min}}}{\sqrt{\pi_{\max}}} (1 - \lambda_1) \|p_Y^1 - p_Y^0\|_2.$

**Comments:** Unlike classical mixing bounds, we seek a lower bound on reconstruction error. This bound is trivial when $\pi_{\min} = 0$, though we can address this with different assumptions (e.g. on the ratio $p_Y^1/p_Y^0$). When $p_Y^1 = p_Y^0$, the lower bound is 0, but this is expected: the reference density already satisfies the constraint and no correction is needed.

When the Markov operator $\mathcal{A}$ is reducible (so that it cannot escape some part of $\mathcal{Y}$ space), the eigenvalue $\lambda_1 = 1$ and again, the lower bound is trivial. A special case of the latter is when $p_{Y|X}$ is a deterministic function from $X$ to $Y$, though in this case, we know that the trivial bound is actually tight, since $p_Y^{det}$ coincides with $p_Y^1$. At the other extreme, if $p_{Y|X}(y|x) > 0$ for all $x, y$ in the support of $p_X^0, p_Y^0$, then $\lambda_1 < 1$, giving a non-zero lower bound on the reconstruction error. More generally, $\lambda_1 < 1$ holds when $\mathcal{A}$ satisfies standard conditions for geometric ergodicity. In this case, we have a non-zero lower bound on the distance from the constraint density, *even when the problem is feasible.*

The takeaway is that practitioners must be careful about applying commonly used schemes from the setting of deterministic transformations to probabilistic ones. Next, we describe a solution to the general problem.

### 3.1. Proposed approach: DualScore

Our approach to solve the general problem is to work with the unconstrained dual problem. We represent the infinite-dimensional dual variable $\phi \in L^2(\mathcal{Y})$ with a neural network $\phi_\theta(y) : \Theta \times \mathcal{Y} \to \mathbb{R}$ with sufficient expressive power. Here, $\theta$ are the neural network parameters, and the dual objective (3), negated from here onwards, becomes:

$$L(\theta) := \left\{ \varepsilon \left( \int_{\mathcal{Y}} \phi_\theta^2(y) \, dy \right)^{\frac{1}{2}} - \int_{\mathcal{Y}} \phi_\theta(y) p_Y^1(y) \, dy \right.$$

$$\left. + \log \int_{\mathcal{X}} p_X^0(x) \cdot \exp \left( \int_{\mathcal{Y}} \phi_\theta(y) p_{Y|X}(y|x) \, dy \right) \, dx \right\}$$

$$:= L_{\mathrm{reg}}(\theta) + L_1(\theta) + L_2(\theta). \quad (5)$$

For any set of parameters $\theta$, define the associated density

$$p_X(x; \theta) = \frac{1}{C_\theta} p_X^0(x) \underbrace{\exp \left( \int_{\mathcal{Y}} \phi_\theta(y) p_{Y|X}(y|x) \, dy \right)}_{w_\theta(x)}. \quad (6)$$

Now, for any $\theta^* \in \{\theta : \arg\min_{\theta \in \Theta} L(\theta)\}$, our approximation to the solution $p_X^*$ of (2) is given by $\widehat{p_X^*}(x) \approx p_X(x; \theta^*)$.

The approximation error of this approach reduces with the expressiveness of the neural network $\phi_\theta(y)$. A natural strategy to optimize the dual is by gradient descent, and we show that it is fairly straightforward to obtain Monte Carlo estimates of the intractable gradient $\nabla_\theta L(\theta)$. This allows a simple and efficient stochastic gradient descent (SGD) algorithm outlined in Algorithm 1. This proceeds by alternating between a simulation step DUALSCOREGENX that generates samples from $p_X(\cdot; \theta^{(k)})$ using the current parameter estimate $\theta^{(k)}$, and an update step that updates $\theta^{(k)}$ to $\theta^{(k+1)}$ by SGD. After optimizing the dual, the same simulation step will be used to generate samples from the density $\widehat{p_X^*}$.

**Estimating** $\nabla_\theta L(\theta) = \nabla_\theta L_{\text{reg}}(\theta) + \nabla_\theta L_1(\theta) + \nabla_\theta L_2(\theta)$. We start with the term $L_1(\theta) = -\mathbb{E}_{y\sim p_Y^1}[\phi_\theta(y)]$: we can directly differentiate under the integral and estimate the gradient with samples $y_{1,1}, \ldots, y_{1,N} \overset{\text{i.i.d.}}{\sim} p_Y^1$:

$$\widehat{\nabla_\theta L_1}(\theta) = -\frac{1}{N}\sum_{i=1}^{N}\nabla_\theta\phi_\theta(y_{1,i}) \approx \nabla_\theta L_1(\theta). \quad (7)$$

For $L_2(\theta) = \log\mathbb{E}_{x\sim p_X^0}\left[\exp\left(\mathbb{E}_{y\sim p_{Y|X}(\cdot|x)}[\phi_\theta(y)]\right)\right]$, we have the following identity (see Appendix C.3):

$$\nabla_\theta L_2(\theta) = \int_\mathcal{X}\int_\mathcal{Y} p_X(x;\theta)p_{Y|X}(y|x)\nabla_\theta\phi_\theta(y)\,dy\,dx. \quad (8)$$

We draw $x_1, \ldots, x_N\sim p_X(\cdot;\theta)$, and for each $x_i$, draw $M$ samples $\{y_{2,ij}\}_{j=1}^M \sim p_{Y|X}(\cdot|x_i)$, with

$$\widehat{\nabla_\theta L_2}(\theta) = \frac{1}{NM}\sum_{i=1}^{N}\sum_{j=1}^{M}\nabla_\theta\phi_\theta(y_{2,ij}). \quad (9)$$

Since $Y$ is typically much lower dimensional than $X$, a good default is $M = 1$ (see Appendix D.1.1).

The final term $L_{\text{reg}}(\theta)$ corresponds to the $\varepsilon$-relaxation of the marginal constraint; we note this disappears when $\varepsilon = 0$. For $\varepsilon > 0$, the following simple importance sampling approach, with $p_Y^1$ as proposal distribution works well:

$$L_{\text{reg}}(\theta) \approx \widehat{L}_{\text{reg}}(\theta) := \varepsilon\left(\frac{1}{N}\sum_{i=1}^{N}\frac{\phi_\theta^2(y_{3,i})}{p_Y^1(y_{3,i})}\right)^{\frac{1}{2}}, \quad y_{3,i}\overset{\text{i.i.d.}}{\sim} p_Y^1.$$

We choose $p_Y^1(y)$ as the proposal distribution since it is easy to sample from, and the term $L_1(\theta)$ in Equation (5) suggests that $p_Y^1(y)$ will be close to $\phi_\theta(y)$ for $\theta$ in the neighborhood of $\theta^*$. Differentiating this Monte Carlo approximation yields

$$\widehat{\nabla_\theta L_{\text{reg}}}(\theta) \approx \varepsilon\nabla_\theta\left(\frac{1}{N}\sum_{i=1}^{N}\frac{\phi_\theta^2(y_{3,i})}{p_Y^1(y_{3,i})}\right)^{\frac{1}{2}}. \quad (10)$$

We compute this gradient directly by automatic differentiation. Note that while $\widehat{L}_{\text{reg}}(\theta)$ is consistent, the square-root

introduces a downward bias in expectation by Jensen's inequality, making the relaxed marginal constraint slightly conservative (see Appendix C.1). This is usually acceptable, as nonzero $\varepsilon$ is typically introduced to relax the problem when feasibility is unknown; when $\varepsilon = 0$, this bias disappears as $L_{\text{reg}}(\theta)$ vanishes. In practice, we can reduce bias using a weighted average across SGD iterates, thus avoiding the need for large Monte Carlo sample size $N$.

The only remaining question is how to sample from the intractable $p_X(\cdot;\theta)$ for any $\theta$. This is needed to estimate $\nabla_\theta L_2(\theta)$, and having found the dual maximizer $\theta^*$, is needed to simulate from the modified density $p_X(x;\theta^*)$.

**DUALSCOREGENX: Sampling from** $p_X(x;\theta)$ **via score correction.** A naive approach extends importance weighting and resampling from the deterministic constraint setting, and following Equation (6), assigns weights $w_\theta(x_i)$ to samples from $x_i\sim p_X^0$ (Tang, 2023). However, the exponential nonlinearity in $w_\theta(\cdot)$ introduces bias and additional variance, and such a scheme does not exploit gradient information we have already calculated while optimizing the dual. More importantly when $p_X^0$ and $p_X^*$ have little overlap (a *density chasm*, Rhodes et al. (2020)), this importance sampling scheme becomes extremely inefficient. Broadly, we cannot expect to explore $p_X^*$ by reweighting samples from $p_X^0$, when the two do not overlap significantly.

Instead, we use the dual neural network to correct the score of the reference density, $\nabla_x\log p_X^0(x)$, which in this work we assume is available. From (6) we have

$$\nabla_x\log p_X(x;\theta) = \nabla_x\log p_X^0(x) + \nabla_x\log w_\theta(x) - \cancel{\nabla_x\log C_\theta}.$$

For the score-correction term $\nabla_x\log w_\theta(x)$, we have

$$\nabla_x\log w_\theta(x) = \mathbb{E}_{y\sim p_{Y|X=x}}\left[\phi_\theta(y)\nabla_x\log p_{Y|X}(y|x)\right] \quad (11)$$

which admits a simple Monte Carlo estimator for any fixed $x \in \mathcal{X}$, provided (once again) $\nabla_x\log p_{Y|X}(y|x)$ is available either in closed-form or via automatic differentiation.

We can now use $\nabla_x\log p_X(x;\theta)$, the score of the corrected distribution to simulate from it. For the general case, we can use SGLD updates (Welling & Teh, 2011):

$$x_i \leftarrow x_{i-1} + \alpha\big(\nabla_x\log p_X^0(x_{i-1}) + \quad (12)$$

$$\frac{1}{J}\sum_{j=1}^{J}\phi_\theta(y_{4,j})\cdot\nabla_x\log p_{Y|X}(y_{4,j}|x_{i-1})\big) + \sqrt{2\alpha}z_i,$$

for $i = 1, \ldots, N$, $z_i \sim N(0, I)$, $y_{4,j}\overset{\text{i.i.d.}}{\sim} p_{Y|X}(\cdot|x_{i-1})$, $J$ is the Monte Carlo batch size, and $\alpha$ is the step size. This produces a Markov chain of samples approximately targeting $p_X(x;\theta)$, allowing us to approximate its expectations. For diffusion models, we correct the scores in the reverse-time step in the same way, allowing us to convert noise into independent samples from the corrected distribution. In

---

**Algorithm 1** Dual-Optimized Score Correction (DUALSCORE)

---

**Training** – Learn $\widehat{\theta}^*$ via **SGD + DUALSCOREGENX**

  **Initialize:** $\theta^{(0)}$ and $x^{(0)} \sim p_X^0$

  **for** $k = 0, \ldots, K_{\text{stop}}$ **do**     {*until stopping criterion*}

    ▷ Produce Monte Carlo samples:

      $\{x_i\}_{i=1}^N \leftarrow \textbf{DUALSCOREGENX}\big(\theta^{(k)}, x^{(k)}\big)$

          {*Input $x^{(k)}$ is not needed for diffusion models*}

      $\{y_{1,i}\}_{i=1}^N, \{y_{3,i}\}_{i=1}^N \sim p_Y^1(\cdot)$

      $\{y_{2,ij}\}_{j=1}^M \sim p_{Y|X}(\cdot \mid x_i), \quad i = 1, \ldots, N$

    ▷ Calculate the estimates $\widehat{\nabla_\theta L_1^{(k)}}, \widehat{\nabla_\theta L_2^{(k)}}, \widehat{\nabla_\theta L_{\text{reg}}^{(k)}}$

       using equations 7, 9 and 10

    ▷ $\widehat{\nabla_\theta L^{(k)}} \leftarrow \widehat{\nabla_\theta L_1^{(k)}} + \widehat{\nabla_\theta L_2^{(k)}} + \widehat{\nabla_\theta L_{\text{reg}}^{(k)}}$

    ▷ $\theta^{(k+1)} \leftarrow \textbf{SGDSTEP}\big(\theta^{(k)}, \widehat{\nabla_\theta L^{(k)}}\big)$

    ▷ $x^{(k+1)} \leftarrow x_N$     {*Not needed for diffusion models*}

  **end for**

---

practice, we update the scores at all time steps of the reverse-diffusion process to avoid abrupt changes in the score (Kim et al., 2024). More details are in the supplementary material.

With this, we can use Equation (8) to obtain a Monte Carlo estimate of the gradient $\widehat{\nabla_\theta L(\theta)}$. We can then take an SGD step to update the neural network parameters $\theta$. Specifically, with a prescribed step-size schedule $\{\eta_k\}_{k \geq 0}$ (we use Adam (Kingma & Ba, 2014) in our implementation), we update the parameters according to SGDSTEP: $\theta^{(k+1)} = \theta^{(k)} - \eta_k \widehat{\nabla_\theta L(\theta^{(k)})}$. Pseudocode is given in Algorithm 1.

We mention one practical consideration for the general SGLD setting: unlike reverse diffusion sampling, which directly generates independent samples from the corrected diffusion model, SGLD produces a correlated Markov chain over $X$. After each update of $\theta$, a naive approach restarts the chain and incurs an additional burn-in cost.

To avoid this inefficiency, we instead maintain a persistent Markov chain that tracks the slowly evolving target distribution $p_X(x; \theta^{(k)})$. After updating $\theta^{(k)}$ to $\theta^{(k+1)}$, we initialize SGLD with the final sample at iteration $k$. When the SGD steps are sufficiently small, the distributions $p_X(\cdot; \theta^{(k)})$ and $p_X(\cdot; \theta^{(k+1)})$ remain close, so only a short additional SGLD run is needed to adapt to the updated target. This persistent-chain strategy requires the SGD learning rate to be small relative to the SGLD mixing time, and is closely related to Persistent Contrastive Divergence (Tieleman, 2008). We discuss the interaction between SGD step size and SGLD mixing further in Section 4.1; see also Section 3.2.

**The density-chasm problem (Rhodes et al., 2020):** Our

SGLD sampling scheme directly explores the support of the target distributions $p_X^*$ without repeatedly reverting to the reference $p_X^0$ as a proposal distribution (as is the case with importance sampling-based approaches). This allows our method to handle situations where a *density chasm* separates the two distributions, a well-known failure mode where low overlap between $p_X^0$ and $p_X^*$ leads to high variance and vanishing effective sample sizes for importance sampling. Even with SGLD, methods that learn the score correction by sampling from $p_X^0$ will estimate it accurately only on $\text{supp}(p_X^0)$; resulting in unreliable gradients in high-probability regions of the target distribution. By contrast, our dual optimization with persistent SGLD concentrates samples on the target $p_X^*$, ensuring the correction is accurate where it is needed. The persistent training procedure also provides a natural initialization mechanism for test-time SGLD sampling: beginning with a sample from $p_X^0$ at the start of training, the chain is gradually annealed toward $p_X^*$ through the sequence of SGD updates, yielding a near-target sample as training progresses. We emphasize that these issues do not arise with the sampler for diffusion models which directly produce independent samples from $p_X^*$.

### 3.2. Related work

**Marginal constraints on generative models.** Existing work imposing marginal constraints has mostly focused on deterministic mappings from $X$ to $Y$, and with $\varepsilon = 0$. In this setting, Kessler et al. (2015) considered Dirichlet process mixture models and introduced a density ratio corrected Metropolis-Hastings acceptance algorithm, while for more general models, Tang (2023) considered both conditional sampling and importance resampling approaches. As we discuss, these are prone to failure in the density-chasm regime. Both Schifeling & Reiter (2016) and Tang & Rao (2025) studied the special case where $f$ is a coordinate projection, enforcing the constraint through synthetic data augmentation or conditional density modeling respectively. Dai et al. (2022) cite the same Altun & Smola (2006) duality result as we do, but develop a specialized approach for deterministic maps, discrete set-data $X$, and binary $Y$.

**Entropic optimal transport and Schrödinger bridge.** The static Schrödinger bridge (SB) problem (Schrödinger, 1932; Marino & Gerolin, 2020) minimizes $\text{KL}(P \| Q)$ over joint distributions $P$ on $\mathcal{X} \times \mathcal{Y}$, where $Q$ is a reference measure, subject to two hard marginal constraints $P_X = \mu$ and $P_Y = \nu$. Closely related is entropy-regularized optimal transport (EOT) (Cuturi, 2013; Benamou et al., 2015). Our problem differs structurally from both. The reference $p_X^0$ lives on $\mathcal{X}$ alone, there is only one marginal constraint (on $Y$), and coupling between $X$ and $Y$ is fixed rather than optimized.

In this context, our dual formulation shares the structure exploited by recent neural EOT methods. Mokrov et al.

(2024) (EgNOT) also use Langevin dynamics to sample from a tilted distribution, and Korotin et al. (2024) (LightSB) avoid MCMC by restricting the dual potential to a quadratic parametric family with a closed-form normalizer. Both methods operate in the two-marginal SB/EOT setting with a fixed reference kernel; our stochastic $p_{Y|X}$ and single-marginal constraint are not covered by any of them.

**Score correction and diffusion model adaptation.** Our score correction $\nabla_x \mathbb{E}_{p_{Y|X}(\cdot|x)}[\phi_\theta(y)]$ is related to classifier guidance (Dhariwal & Nichol, 2021), which modifies the diffusion score by $\nabla_x \log p(c|x_t)$ to steer generation toward a fixed class label. For our scheme, the target is a population-level constraint rather than a fixed class, and is learned via dual optimization rather than provided by a pre-trained classifier. This last distinction extends to a broader family of inference-time score modification methods, including reward and energy guidance (Bansal et al., 2024; Guo et al., 2024), plug-and-play likelihood corrections (Graikos et al., 2022; Meng & Kabashima, 2025), and entropy-regularized control (Uehara et al., 2024).

A related body of work uses the score function to sample from the diffusion posterior $p_X^0(x|y)$ under a known measurement model $y = Hx + \varepsilon$ (Song et al., 2022; Chung et al., 2022; 2023), while the Twisted Diffusion Sampler (Wu et al., 2023) replaces these approximations with sequential Monte Carlo. These methods target a single observation $y$; our problem instead enforces a distributional constraint on the marginal of $Y$ across the generated population.

Similar in spirit to our work is Khalafi et al. (2024), who also formulate constrained generation as a dual optimization problem. Their constraints control the KL divergence between the model and $m$ desired data distributions, with the latter in the $\mathcal{X}$-space rather than in the $\mathcal{Y}$-space. The optimal solution is a finite mixture of the original and constraint distributions, with a finite-dimensional dual variable controlling the mixture weights. In Kim et al. (2023) and Kim et al. (2024) diffusion model scores are corrected by learning the time-dependent density ratio $p_t^1(x_t)/p_t^0(x_t)$ via a binary classifier, using samples from both $p_X^0$ and a target distribution $p_X^1$, whereas we require only specification of the constraint marginal $p_Y^1$. Our time-dependent extension of the constrained problem (Appendix E) is directly inspired by their observation that score correction applied along the full diffusion time axis avoids abrupt score discontinuities.

**SGD and two-timescale optimization.** Our training algorithm maintains a persistent Markov chain for $X$ across SGD iterations, initializing each SGLD chain with the final state of the previous iteration. This relates to Persistent Contrastive Divergence (PCD) (Tieleman, 2008; Tieleman & Hinton, 2009), introduced for training restricted Boltzmann machines. PCD reduces burn-in cost but introduces bias when the parameter update scale is large relative to the mix-

ing rate of the SGLD dynamics, leading to samples that lag behind the current model. Similar ideas arise in reinforcement learning: on-policy actor–critic methods (Konda & Tsitsiklis, 2003; Schulman et al., 2017) rely on trajectories generated from the current policy, which remain approximately valid under small policy updates, whereas off-policy actor–critic methods (Fujimoto et al., 2018; Haarnoja et al., 2018) use experience replay to reuse past transitions. More broadly, these approaches can be viewed through the lens of two-timescale stochastic approximation (Borkar, 1997; Konda & Tsitsiklis, 2004), where a fast mixing Markov chain tracks a target distribution induced by slowly evolving parameters. This has been studied theoretically in Oliva et al. (2025), and we empirically study the interaction between the SGD and SGLD step sizes in Appendix D.1.2.

## 4. Experiments

For synthetic and NPMLE tasks, we use a two-hidden-layer MLP with sizes (32, 16) and ReLU activations to parameterize $\phi : \mathcal{Y} \to \mathbb{R}$. We chose Monte Carlo batch sizes from $\{16, 32, 64\}$ and learning rates from $\{10^{-5}, 10^{-4}, 10^{-3}\}$. For image tasks, we use the extended time-dependent dual variable described in Section 4.3, parameterized as a time-dependent MLP $\phi : [0, T] \to \mathbb{R}^{|\mathcal{Y}|}$, with three hidden layers of size (32, 32, 32) and the same hyperparameter ranges. Details on training and hyperparameter tuning not described below can be found in Appendix D.2.

### 4.1. Synthetic Tasks

We start with a simple setup: the reference $p_X^0$ is Gaussian, and $Y$ is a linear transformation of $X$ plus additive noise. We consider Gaussian and Laplace noise. For clarity, we present results for the 1-d setting with $f(x) = 2x + 1$, $\sigma_{Y|X} = 1$ and $p_Y^1 = \mathcal{N}(-1, 2^2)$. This problem is feasible, with solution $p_X^* = \mathcal{N}(-1, (3/4)^2)$ when $p_{Y|X}$ is Gaussian (see Appendix D.2.1). We vary $p_X^0 \in \{\mathcal{N}(0, 1^2), \mathcal{N}(-10, 2^2)\}$, the latter inducing a severe density chasm between $p_X^0$ and $p_X^*$.

We evaluate any candidate solution $\widehat{p_X^*}$ along two metrics: (i) *constraint gap*, $\|\widehat{p_Y^*} - p_Y^1\|_2$: for $\varepsilon = 0$, smaller is better; for $\varepsilon > 0$, we check how closely $\|\widehat{p_Y^*} - p_Y^1\|_2 \leq \varepsilon$ is satisfied; (ii) *distributional shift*, $D_{\text{KL}}(\widehat{p_X^*}\|p_X^0)$: for the same constraint gap, smaller is better. In addition to the baselines CONDSMPL and IRS after Corollary 3.2, we include a baseline: SGD+IRS. This optimizes the dual by SGD, but samples from $p_X(\cdot; \theta)$ using importance resampling (IRS) with weights $w_\theta(x)$ in Equation (6) (rather than SGLD).

Figure 1 summarizes the results. Across all settings, all methods reduce the constraint gap $\|p_Y^0 - p_Y^1\|_2$ associated with the reference model $p_X^0$. When no density chasm is

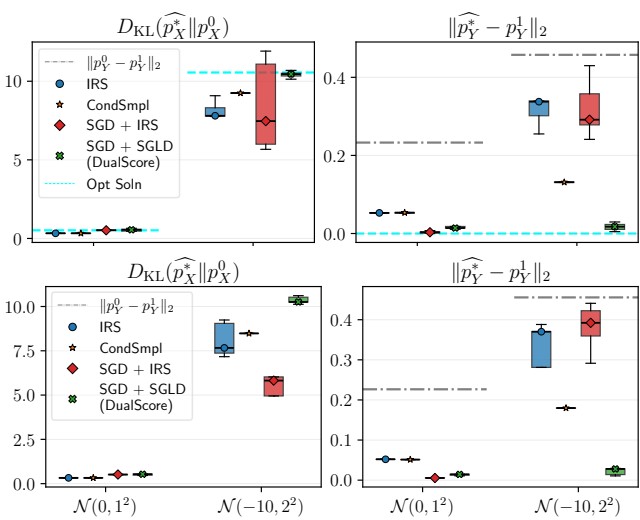

*Figure 1.* Synthetic results: (top) Gaussian (bottom) Laplace noise

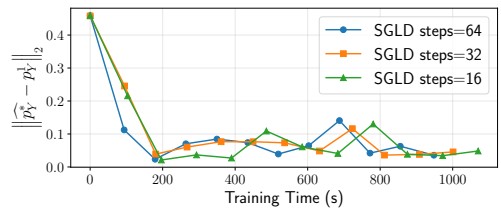

*Figure 2.* Constraint gap vs epoch for different SGLD step sizes.

present, methods designed for probabilistic constraints consistently achieve smaller constraint gaps than those for deterministic transformations. Additionally, with a density chasm, importance-resampling-based approaches become ineffective due to the fixed misspecified proposal distribution. Our proposed SGLD approach remains effective in this regime, successfully reducing the constraint gap across all scenarios. This story is repeated in synthetic experiments with higher dimension; we include these in Appendix D.1.3.

Figure 2 shows the evolution of the constraint gap over SGD training for different numbers of SGLD steps per iteration (i.e., the Monte Carlo size $N$ in Algorithm 1). The results are fairly stable even with as few as 16 SGLD steps per iteration, with relatively short chains giving reliable gradient estimates. We further examine the interaction between SGLD and SGD step sizes in Appendix D.1.2, where we find that PCD-style instability from stale samples does not appear unless the SGD learning rate is too large.

### 4.2. Regularized nonparametric MLE (NPMLE)

We consider the NPMLE task outlined in Section 1, P2. Write $p_Y^1$ for the smoothed empirical density of i.i.d. observations $\{y_i\}_{i=1}^N$ (we use a kernel density estimate (KDE)). There are obtained by perturbing (through a known condi-

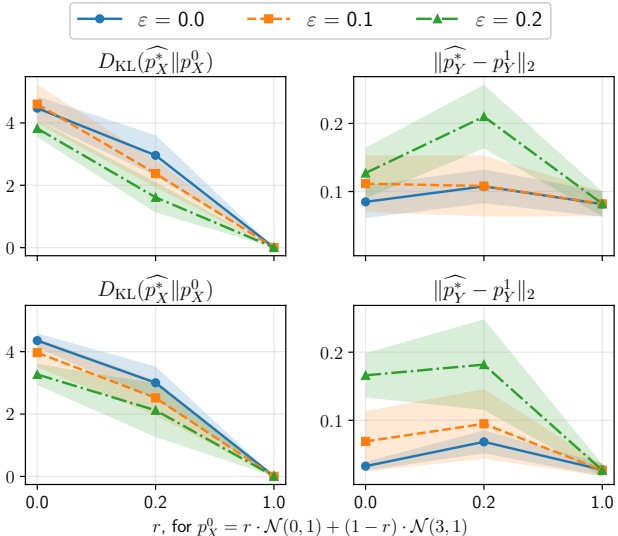

*Figure 3.* NPMLE results of DUALSCORE for dataset sizes 100 (top) and 100,000 (bottom). Increasing $r$ along the x-axis **decreases** the density chasm between $p_X^0$ and $p_Y^1$. Lines show means over 5 runs, and shaded ribbons indicate one standard deviation.

tional $p_{Y|X}$) another i.i.d. dataset $\{x_i\}_{i=1}^N$ from an unknown density $p_X^T$. Our NPMLE estimate $p_X^*$ of $p_X^T$ is regularized towards a density $p_X^0$. Then, $p_X^*$ is obtained as the solution of our constrained optimization problem.

We use the following settings: $p_X^0 = \mathcal{N}(0, 1)$, $p_{Y|X}(\cdot|x) = \mathcal{N}(x, 0.2^2)$, $p_Y^1 = \text{KDE}\left(\{y_i\}_{i=1}^N\right)$, $p_X^T \equiv r\mathcal{N}(0, 1) + (1 - r)\mathcal{N}(3, 1)$ for $r \in \{0, 0.2, 1\}$. Thus, $p_X^T$ is a two-component Gaussian mixture, with one component equal to the prior $p_X^0$. The mixture weight $r$ controls the deviation from $p_X^0$, and the density chasm. We consider $\varepsilon$ in $\{0, 0.1, 0.2\}$; in practice, this will be determined by the sample size $S$, with larger $S$ requiring stronger constraints as we seek to penalize deviations from the empirical estimate more heavily.

Figure 3 shows results when the sample size $S = 100$ and $100,000$. In most cases, the constraint is satisfied, with the returned solution close to the constraint boundary. In other words, as we would hope, our approach deviates from the target density to the maximum extent permitted, in order to minimize the KL divergence from the reference. Additionally, enforcing the constraint more rigidly results in solutions further away from the reference density.

There are two exceptions to this. The first is for small $r$ and $\varepsilon = 0$ (the hardest setting). Now, the constraint is violated, a consequence of the problem being infeasible. However, the solution obtained by terminating our dual optimization scheme is still a reasonable one, having $L^2$ distance of about 0.1 from the target marginal (instead of the impossible 0). An implication is that the feasibility threshold $\varepsilon_* \approx 0.1$.

The second exception is when $r = 1$: now the reference

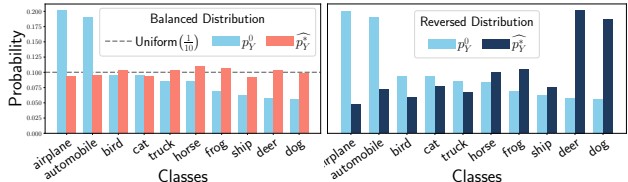

*Figure 4.* CIFAR-10(LT/5%) class frequencies before/after training

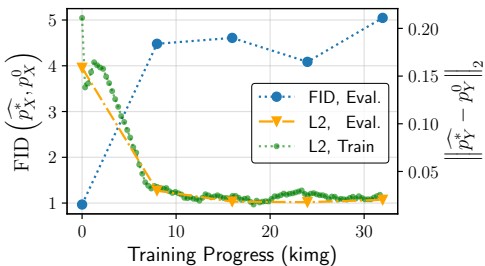

*Figure 5.* Training dynamics on CIFAR-10 (LT/5%). The training $L^2$ decreases smoothly and exhibits stable convergence. We track a moving average of $L^2$ (window size: 5K images). The training $L^2$ distance reaches its minimum at approximately 18K images.

$p_X^0$ is already very close to the target $p_X^*$ before any score correction. As a result all settings of $\varepsilon$ return solutions with low $L^2$ distance from the target density. We see this even more strongly with the larger sample size, when the KDE estimate is closer to the true marginal.

### 4.3. Image Generation with Class-level Constraints

For this task, $p_X^0$ corresponds to a pretrained score-based diffusion model (Song et al., 2021; Karras et al., 2022) associated with a 'time'-dependent parameterized score function $s_\eta(\boldsymbol{x}_t, t)$ (with $\nabla_{\boldsymbol{x}_0} \log p_X^0(\boldsymbol{x}_0) = s_\eta(\boldsymbol{x}_0, 0)$). This, together with an attribute classifier $p_{Y|X}(y|x)$ implies a distribution over classes $p_Y^0$. We consider two target marginal distributions $p_Y^1$: (i) **Balanced distribution**: the standard setting in the literature (Choi et al., 2020; Kim et al., 2024), where the target marginal $p_Y^1$ is uniform over classes, and (ii) **Reversed distribution**: where $p_Y^1$ is constructed by reversing the rank order (by frequency) of classes in $p_Y^0$.

We extend our methodology to exploit the additional gradient structure provided by diffusion models. Following Kim et al. (2024), we apply marginal constraint to the diffusion model densities at all times (recall $t = 1$ corresponds to noise that the diffusion model converts to images at $t = 0$). We allow the constraint $\varepsilon(t)$ to vary with diffusion time, so that it is applied more strongly from $t = 1$ to $t = 0$ as the diffusion model converts noise to signal, with $\varepsilon(0)$ equal to the desired constraint parameter $\varepsilon$. From this, and with $\lambda(t)$ the temporal weighting function of the diffusion model, we construct the time-weighted dual as $L(\theta) = \mathbb{E}_{t \sim \mathcal{U}[0,T]}[\lambda(t) L(\theta, t)]$ where

$$
L(\theta, t) = \left\{ \varepsilon(t) \left( \int_{\mathcal{Y}} \phi_\theta^2(y, t)\, dy \right)^{\frac{1}{2}} - \int_{\mathcal{Y}} \phi_\theta(y, t) p_Y^1(y)\, dy + \right.
$$
$$
\left. \log \int_{\mathcal{X}} p_{X(t)}^0(\boldsymbol{x}_t) \cdot \exp \left( \int_{\mathcal{Y}} \phi_\theta(y, t) p_{Y|X(t)}(y|x_t)\, dy \right) dx_t \right\}
$$
$$
:= L_{\text{reg}}(\theta, t) + L_1(\theta, t) + L_2(\theta, t). \tag{13}
$$

The optimization scheme described in Section 3.1 remains mostly unchanged, with the diffusion sampler having the adapted score function $s_\eta(\boldsymbol{x}_t, t) + \nabla_{x_t} \log w_\theta(x_t, t)$. For more details, please refer to Appendix E.1.

**Datasets** We use CIFAR-10 (Krizhevsky et al., 2009) and CelebA (Liu et al., 2015), following the fair-generation setups of Kim et al. (2024). For CIFAR-10, we evaluate all long-tailed settings LT/$q\%$, $q \in \{5, 10, 25, 50\}$, where the training set combines a long-tailed biased subset $\mathcal{D}_{\text{bias}}$ (Cao et al., 2019) with a small uniformly subsampled unbiased subset $\mathcal{D}_{\text{fair}}$ satisfying $|\mathcal{D}_{\text{fair}}|/|\mathcal{D}_{\text{bias}}| = q\%$. For CelebA, we use the original training set, which is naturally biased with respect to the joint attribute of gender and hair color.

**Setup** For each dataset, we train an unconditional diffusion model using the EDM framework (Karras et al., 2022), yielding a time-dependent score function $s_\eta(\boldsymbol{x}_t, t) = \nabla_{\boldsymbol{x}_t} \log p_{X(t)}^0(\boldsymbol{x}_t)$. As Figure 4 shows, generated samples inherit class imbalances present in the training data. We then follow Kim et al. (2024; 2023) and train time-dependent classifiers on noisy diffusion states $X(t)$ to obtain the time-dependent conditional $p_{Y|X(t)}$. The classifier uses a fixed ADM feature extractor (Dhariwal & Nichol, 2021) followed by a shallow trainable U-Net encoder (Ronneberger et al., 2015) with a softmax output over target classes.

We then optimize our proposed time-weighted dual objective to learn $\phi_\theta$. Since $Y$ is discrete, $\phi_\theta(\cdot, t) : \mathcal{T} \to \mathbb{R}^{|\mathcal{Y}|}$ depends only on diffusion time and is parameterized by a lightweight multilayer perceptron. Additional implementation details are provided in Appendix D.2.

As is standard, we report $\text{FID}(\widehat{p_X^*}, p_X^0)$, the Fréchet Inception Distance (Heusel et al., 2017) between the original and corrected models as a practical proxy for their KL divergence (estimated using 50,000 samples like Kim et al. (2024)). To evaluate the effect of classifier stochasticity, we inject uniform noise at level $\rho \in \{0, 0.25, 0.5, 0.75\}$ into the classifier, so that the actual label distribution at any time $t$ follows $\widetilde{p}_{Y|X(t)} = (1 - \rho)p_{Y|X(t)} + \rho p_U$, where $p_U$ is uniform over classes. As before, we quantify constraint satisfaction by $\|\widehat{p_Y^*} - p_Y^1\|_2$.

We compare with IRS and CONDSMPL, two baselines derived for deterministic constraints (Section 3). CONDSMPL corresponds to the conditional sub-sampling strategy used

| Task | Dataset | $\widehat{p^*_X}$ | FID $\left(\widehat{p^*_X}, p^0_X\right)$ (as noise-level $\rho$ increases) | | | | $\left\|\widehat{p^*_Y} - p^1_Y\right\|_2$ (as noise-level $\rho$ increases) | | | |
|---|---|---|---|---|---|---|---|---|---|---|
| | | | $\rho = 0$ | $\rho = 0.25$ | $\rho = 0.5$ | $\rho = 0.75$ | $\rho = 0$ | $\rho = 0.25$ | $\rho = 0.5$ | $\rho = 0.75$ |
| Balanced | CelebA | $p^0_X$ | 0.457 | - | - | - | 0.226 | 0.168 | 0.113 | 0.056 |
| | | IRS | 1.399 | 0.896 | 0.706 | 0.668 | 0.017 | 0.088 | 0.090 | 0.054 |
| | | CONDSMPL | 1.415 | 0.925 | 0.719 | 0.686 | 0.019 | 0.088 | 0.091 | 0.054 |
| | | DUALSCORE | 1.159 | 1.209 | 1.049 | 1.089 | **0.011** | **0.010** | **0.009** | **0.003** |
| | CIFAR-10 | $p^0_X$ | 0.947 | - | - | - | 0.158 | 0.117 | 0.079 | 0.039 |
| | (LT/5%) | IRS | 4.537 | 2.602 | 1.658 | 1.418 | 0.009 | 0.053 | 0.058 | 0.037 |
| | | CONDSMPL | 4.665 | 2.601 | 1.713 | 1.440 | **0.007** | 0.053 | 0.057 | 0.037 |
| | | DUALSCORE | 3.936 | 4.347 | 3.957 | 4.023 | 0.014 | **0.013** | **0.007** | **0.004** |
| Reversed | CelebA | $p^0_X$ | 0.457 | - | - | - | 0.444 | 0.388 | 0.332 | 0.277 |
| | | IRS | 3.395 | 1.611 | 0.975 | 0.711 | 0.040 | 0.207 | 0.269 | 0.264 |
| | | CONDSMPL | 3.416 | 1.738 | 1.078 | 0.781 | 0.040 | 0.207 | 0.266 | 0.264 |
| | | DUALSCORE | 3.094 | 5.100 | 61.213 | 127.591 | **0.038** | **0.059** | **0.031** | **0.097** |
| | CIFAR-10 | $p^0_X$ | 0.947 | - | - | - | 0.283 | 0.249 | 0.214 | 0.185 |
| | (LT/5%) | IRS | 12.819 | 5.704 | 2.984 | 1.733 | **0.019** | 0.127 | 0.168 | 0.176 |
| | | CONDSMPL | 12.864 | 6.136 | 3.051 | 1.882 | 0.021 | 0.125 | 0.171 | 0.176 |
| | | DUALSCORE | 12.713 | 17.850 | 40.707 | 113.409 | 0.046 | **0.040** | **0.038** | **0.041** |

*Table 1.* **Our method effectively reduces probabilistic constraint violation and remains robust as the classifier transformation becomes noisier.** We report $\mathrm{FID}(\widehat{p^*_X}, p^0_X)$, measuring fidelity to the original model, and $\|\widehat{p^*_Y} - p^1_Y\|_2$, measuring marginal constraint violation. The case $\rho = 0$ corresponds to the noise-free classifier; larger $\rho$ indicates stronger uniform noise injected into the classifier transformation. The baseline $\mathrm{FID}(p^0_X, p^0_X)$ is estimated by resampling from the original generator, reflecting intrinsic sampling variability (expected $\approx 0$). Complete results are provided in Appendix D.1.5, including additional CIFAR-10 settings LT/$q$% with $q \in \{10, 25, 50\}$.

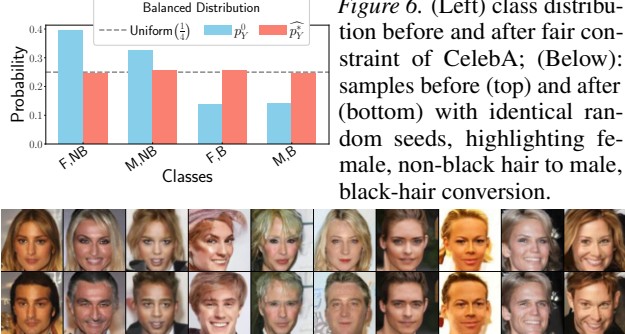

*Figure 6.* (Left) class distribution before and after fair constraint of CelebA; (Below): samples before (top) and after (bottom) with identical random seeds, highlighting female, non-black hair to male, black-hair conversion.

in prior fair-generation approaches to construct target fair distributions (Choi et al., 2020; Kim et al., 2024).

**Results.** Table 1 reports quantitative results. Across tasks, datasets and noise levels, our method consistently reduces the marginal constraint gap $\|p^0_Y - p^1_Y\|_2$ associated with the biased generator $p^0_X$, while keeping the corrected generator relatively close to $p^0_X$ w.r.t. FID. In particular, we achieve near-uniform class distributions in the balanced task and accurately match the reversed target distribution (see Figures 4 and 6), confirming the flexibility of our framework beyond fairness. The comparison with IRS and CONDSMPL highlights the difference between deterministic and probabilistic constraint handling. When $\rho = 0$, when the classifier transformation is unperturbed, these methods sometimes achieve slightly lower $L^2$ discrepancy (though neither consistently outperforms ours). As $\rho$ increases however, IRS and CONDSMPL degrade noticeably, whereas our method continues to achieve substantially lower $L^2$ discrepancy, with the larger FID reflecting the greater distributional shift required by the constrained solution.

## 5. Discussion

We introduced DUALSCORE, a framework for correcting the score function of a score-based model to satisfy side knowledge in the form of marginal constraints on stochastic transformations of the modeled variable. We showed that naive importance reweighting and conditional sampling fail in this general setting, and that the dual of the constrained KL-minimization problem yields an additive score correction that we can learn and simulate from without modifying the original model.

Future work includes extending our framework to more complex sample spaces, including discrete spaces where the score is not naturally defined, infinite-dimensional spaces corresponding to stochastic process models, and structured spaces such as protein structures, trees, and trajectories. We consider only a single marginal constraint; practical settings often involve knowledge about multiple aspects of the modeled objects, possibly mutually inconsistent. Developing principled approaches to the multi-constraint setting is an important next step. Our formulation uses KL divergence from the reference and an $L^2$ norm for the relaxed marginal constraint; alternative divergence measures and constraint norms may yield dual objectives whose gradients are easier to estimate or better suited to specific applications. Finally, a more rigorous theoretical analysis of the errors introduced by neural network approximation, Monte Carlo estimation, SGD convergence, and SGLD approximation and mixing remains an important open direction.

**Software:** Available at https://github.com/wushishang/dual-score

**Acknowledgements** VR acknowledges support from the National Science Foundation through grant DMS-2503118.

## Impact Statement

Score-based generative models are increasingly deployed in a number of critical domains. This work enables practitioners to enforce distributional constraints on pre-trained models, for example, correcting demographic imbalances in generated content or ensuring consistency with domain knowledge, without retraining. The primary intended benefit is improved alignment between model outputs and societal or scientific requirements. However, the same mechanism could be misused to deliberately steer generated content to amplify rather than correct biases.

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

# Supplementary Material for
# Generative modeling with probabilistic constraints

## A. Infeasibility under stochastic transformations

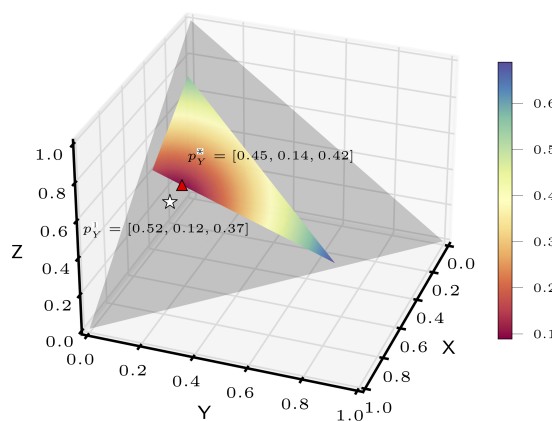

*Figure 7.* Solutions may fail to exist in the stochastic transformation setting when $\varepsilon$ is small. For a given target marginal $p_Y^1 \in \mathcal{P}_3$ (shown as a star), there exists a stochastic transition matrix $P_{Y|X} \in \mathbb{R}^{3 \times 3}$ such that $\inf_{p_X \in \mathcal{P}_3} \left\| p_Y^1 - P_{Y|X} \circ p_X \right\|_2 > 0$, where $\mathcal{P}_3$ denotes the three-dimensional probability simplex. The grey region represents $\mathcal{P}_3$, while the colored surface corresponds to $\{P_{Y|X} \circ p_X : p_X \in \mathcal{P}_3\}$, with the colormap indicating the associated $L_2$ distance to $p_Y^1$. In particular, the red triangle marks the point $p_Y^*$ that empirically minimizes the $L_2$ distance via grid search.

## B. Proofs

Proposition 3.3 from the main text is labeled as Proposition B.2. Throughout this appendix, all conditional expectations $\mathbb{E}$ are under the joint $p_{Y|X}(y|x)p_X^0(x)$.

**Lemma B.1.** *Define the Markov operator* $\mathcal{A} : L^2(p_Y^0) \to L^2(p_Y^0)$ *as* $(\mathcal{A}f)(y) := \mathbb{E}[\mathbb{E}[f(Y')|X]|Y = y] = \int \left( \int p_{Y|X}(y'|x) f(y') \, dy' \right) \frac{p_{Y|X}(y|x)p_X^0(x)}{p_Y^0(y)} \, dx$. *Then, this operator is self-adjoint and positive semidefinite on* $L^2(p_Y^0)$, *with operator norm* $\leq 1$ *and with largest eigenvalue equal to* $1$.

*Proof.* Most of these properties are a result of $\mathcal{A}$ being a reversible Markov operator with invariant measure $p_Y^0$. For completeness, we include a proof.

Define $f = p_Y^1/p_Y^0 \in L^2(p_Y^0)$, and let $\phi$ be any element of $L^2(p_X^0)$. Let $1_X \in L^2(p_X^0)$ and $1_Y \in L^2(p_Y^0)$ denote the constant-one functions on the supports of $p_X^0$ and $p_Y^0$, respectively.

Consider the normalized forward operator $\tilde{T} : L^2(p_X^0) \to L^2(p_Y^0)$,

$$(\tilde{T}\phi)(y) := \mathbb{E}[\phi(X) \mid Y = y] = \frac{1}{p_Y^0(y)} \int p_{Y|X}(y \mid x) \, \phi(x) \, p_X^0(x) \, dx,$$

defined for $p_Y^0$-a.e. $y$. Its adjoint $\tilde{T}^\dagger : L^2(p_Y^0) \to L^2(p_X^0)$ is defined by

$$\langle \tilde{T}\phi, f \rangle_{L^2(p_Y^0)} = \langle \phi, \tilde{T}^\dagger f \rangle_{L^2(p_X^0)} \qquad \text{for all } \phi \in L^2(p_X^0), \ f \in L^2(p_Y^0).$$

Indeed,

$$\langle \tilde{T}\phi, f \rangle_{L^2(p_Y^0)} = \int \frac{1}{p_Y^0(y)} \left( \int p_{Y|X}(y \mid x) \, \phi(x) \, p_X^0(x) \, dx \right) f(y) \, p_Y^0(y) \, dy$$

$$= \int \phi(x) \left( \int p_{Y|X}(y \mid x) \, f(y) \, dy \right) p_X^0(x) \, dx.$$

Therefore, $(\tilde{T}^\dagger f)(x) = \int p_{Y|X}(y \mid x) \, f(y) \, dy = \mathbb{E}[f(Y) \mid X = x]$.

Hence, for any $f \in L^2(p_Y^0)$, $(\tilde{T}\tilde{T}^\dagger f)(y) = \mathbb{E}[\mathbb{E}[f(Y') \mid X] \mid Y = y]$, implying $\mathcal{A} = \tilde{T}\tilde{T}^\dagger$.

Note that $\tilde{T}\mathbf{1}_X = \mathbf{1}_Y$ and $\tilde{T}^\dagger \mathbf{1}_Y = \mathbf{1}_X$, we have that $\mathbf{1}_Y$ is a fixed point of $\mathcal{A} = \tilde{T}\tilde{T}^\dagger$. Therefore, $\mathbf{1}_Y$ is an eigenfunction of $\mathcal{A}$ associated with eigenvalue 1.

*Self-adjointness:* For any $f, g \in L^2(p_Y^0)$, using the adjoint relation $\langle \tilde{T}\phi, f \rangle_{L^2(p_Y^0)} = \langle \phi, \tilde{T}^\dagger f \rangle_{L^2(p_X^0)}$:
$\langle \tilde{T}\tilde{T}^\dagger f, g \rangle_{L^2(p_Y^0)} = \langle \tilde{T}^\dagger f, \tilde{T}^\dagger g \rangle_{L^2(p_X^0)} = \langle f, \tilde{T}\tilde{T}^\dagger g \rangle_{L^2(p_Y^0)}$. So $\tilde{T}\tilde{T}^\dagger$ is self-adjoint on $L^2(p_Y^0)$.

*Positive semidefiniteness:* For any $f \in L^2(p_Y^0)$: $\langle \tilde{T}\tilde{T}^\dagger f, f \rangle_{L^2(p_Y^0)} = \langle \tilde{T}^\dagger f, \tilde{T}^\dagger f \rangle_{L^2(p_X^0)} = \|\tilde{T}^\dagger f\|_{L^2(p_X^0)}^2 \geq 0$.

*Operator norm $\leq 1$:* Since $\|\tilde{T}\|_{L^2(p_X^0)} = \|\tilde{T}^\dagger\|_{L^2(p_Y^0)}$, we have $\|\mathcal{A}\|_{L^2(p_Y^0)} \leq \|\tilde{T}\|_{L^2(p_X^0)}\|\tilde{T}^\dagger\|_{L^2(p_Y^0)} = \|\tilde{T}^\dagger\|_{L^2(p_Y^0)}^2$. Thus it suffices to show $\|\tilde{T}^\dagger f\|_{L^2(p_X^0)} \leq \|f\|_{L^2(p_Y^0)}$ for all $f \in L^2(p_Y^0)$.

Now recall that $(\tilde{T}^\dagger f)(x) = \int p_{Y|X}(y|x) \, f(y) \, dy$. By Jensen's inequality, $\left( \int p_{Y|X}(y|x) \, f(y) \, dy \right)^2 \leq \int p_{Y|X}(y|x) \, f(y)^2 \, dy$. Hence $\|\tilde{T}^\dagger f\|_{L^2(p_X^0)}^2 = \int \left( \int p_{Y|X}(y|x) \, f(y) \, dy \right)^2 p_X^0(x) \, dx \leq \int \left( \int p_{Y|X}(y|x) \, f(y)^2 \, dy \right) p_X^0(x) \, dx = \int f(y)^2 \left( \int p_{Y|X}(y|x) \, p_X^0(x) \, dx \right) dy = \int f(y)^2 p_Y^0(y) \, dy = \|f\|_{L^2(p_Y^0)}^2$. $\square$

**Proposition B.2.** *Define the Markov operator* $\mathcal{A} : L^2(p_Y^0) \to L^2(p_Y^0)$ *as* $(\mathcal{A}f)(y) := \mathbb{E}[\mathbb{E}[f(Y'|X)]|Y = y] = \int \left( \int p_{Y|X}(y'|x) \, f(y') \, dy' \right) \frac{p_{Y|X}(y|x)p_X^0(x)}{p_Y^0(y)} \, dx$. *Then:*
*a)* $\mathcal{A}$ *has eigenvalues in* $[0, 1]$ *with largest eigenvalue* 1.
*b) Let* $\lambda_1$ *denote its second largest eigenvalue, so that* $1 - \lambda_1$ *is the spectral gap. Let* $\pi_{\min} \leq p_Y^0(y) \leq \pi_{\max} \, \forall y \in \operatorname{supp}(p_Y^0)$. *Then we can bound the induced marginal* $p_Y^{det}$ *of the deterministic scheme away from the target* $p_Y^1$ *as*
$$\|p_Y^1 - p_Y^{det}\|_2 \geq \frac{\sqrt{\pi_{\min}}}{\sqrt{\pi_{\max}}} (1 - \lambda_1) \|p_Y^1 - p_Y^0\|_2.$$

*Proof.* Define $f = p_Y^1/p_Y^0 \in L^2(p_Y^0)$. The posterior marginal is: $p_X^{det}(x) = \int \frac{p_{Y|X}(y|x) \, p_X^0(x)}{p_Y^0(y)} p_Y^1(y) \, dy = p_X^0(x) \cdot \left( \tilde{T}^\dagger f \right)(x)$. The reconstructed marginal of $Y$ is:

$$p_Y^{det}(y) = \int p_{Y|X}(y|x) \, p_X^{det}(x) \, dx = \int p_{Y|X}(y|x) \, p_X^0(x) \left( \tilde{T}^\dagger f \right)(x) \, dx = p_Y^0(y) \cdot (\tilde{T}\tilde{T}^\dagger f)(y).$$

Therefore: $\frac{p_Y^{det}}{p_Y^0} = \mathcal{A}f$, and the distance of interest is: $\|f - \mathcal{A}f\|_{L^2(p_Y^0)} = \|(I - \mathcal{A})f\|_{L^2(p_Y^0)}$.

Decompose: $f = \mathbf{1}_Y + f_\perp$, $f_\perp = \frac{p_Y^1 - p_Y^0}{p_Y^0}$. Note that $\langle f_\perp, \mathbf{1}_Y \rangle_{L^2(p_Y^0)} = 0$ since $p_Y^1$ and $p_Y^0$ integrate to 1. Since $\mathcal{A}\mathbf{1}_Y = \mathbf{1}_Y$:

$$\frac{p_Y^1 - p_Y^{det}}{p_Y^0} = f - \mathcal{A}f = (I - \mathcal{A})f_\perp.$$

Let $\lambda_1$ be the second largest eigenvalue of $\mathcal{A}$ on $L^2(p_Y^0)$, so that the spectral gap is $1 - \lambda_1$. By the spectral theorem, on the subspace orthogonal to $\mathbf{1}_Y$ in $L^2(p_Y^0)$,

$$\|(I - \mathcal{A})f_\perp\|_{L^2(p_Y^0)} \geq (1 - \lambda_1)\|f_\perp\|_{L^2(p_Y^0)}.$$

Substituting

$$\left\| \frac{p_Y^1 - p_Y^{det}}{p_Y^0} \right\|_{L^2(p_Y^0)} \geq (1 - \lambda_1) \left\| \frac{p_Y^1 - p_Y^0}{p_Y^0} \right\|_{L^2(p_Y^0)}.$$

To control the raw $L^2$ norm, we need addition tail conditions on $p_1^Y/p_0^Y$. For example, assume $\pi_{\min} \leq p_Y^0(y) \leq \pi_{\max}$. Then we have:

$$\frac{1}{\sqrt{\pi_{\min}}}\|p_Y^1 - p_Y^{det}\|_{L^2} \geq (1 - \lambda_1) \cdot \frac{1}{\sqrt{\pi_{\max}}}\|p_Y^1 - p_Y^0\|_{L^2} \quad \text{which gives:}$$

$$\|p_Y^1 - p_Y^{det}\|_{L^2} \geq \frac{\sqrt{\pi_{\min}}}{\sqrt{\pi_{\max}}}(1 - \lambda_1)\|p_Y^1 - p_Y^0\|_{L^2}.$$

$\square$

# C. Bias, Consistency, and Complexity Analysis

## C.1. Bias and Consistency of the Square-root Regularization Estimator

**Proposition C.1.** *Let*

$$Z_i := \frac{\phi_\theta^2(Y_i)}{p_Y^1(Y_i)}, \qquad Y_1, \ldots, Y_N \overset{i.i.d.}{\sim} p_Y^1,$$

*and define*

$$\mu_\theta := \mathbb{E}[Z_1], \qquad \sigma_\theta^2 := \mathrm{Var}(Z_1), \qquad L_{\mathrm{reg}}(\theta) := \sqrt{\mu_\theta}, \qquad \widehat{L}_{\mathrm{reg}}(\theta) := \sqrt{\bar{Z}_N},$$

*where*

$$\bar{Z}_N := \frac{1}{N}\sum_{i=1}^{N} Z_i.$$

*Assume that $\mu_\theta > 0$ and $\sigma_\theta^2 < \infty$. Then:*

1. *$\mathbb{E}[\widehat{L}_{\mathrm{reg}}(\theta)] \leq L_{\mathrm{reg}}(\theta)$.*

2. *$\widehat{L}_{\mathrm{reg}}(\theta) \to L_{\mathrm{reg}}(\theta)$ almost surely.*

3. *$\widehat{L}_{\mathrm{reg}}(\theta) - L_{\mathrm{reg}}(\theta) = O_p(N^{-1/2})$.*

*Proof.* We write $g(x) := \sqrt{x}$, so that

$$\widehat{L}_{\mathrm{reg}}(\theta) = g(\bar{Z}_N), \qquad L_{\mathrm{reg}}(\theta) = g(\mu_\theta).$$

**Downward bias.** Since $g$ is concave on $\mathbb{R}_+$, by Jensen's inequality,

$$\mathbb{E}[\widehat{L}_{\mathrm{reg}}(\theta)] \leq g\left(\mathbb{E}[\bar{Z}_N]\right) = g(\mu_\theta) = L_{\mathrm{reg}}(\theta).$$

**Consistency.** Since $Z_1, \ldots, Z_N$ are i.i.d. with $\mathbb{E}[|Z_1|] < \infty$, the strong law of large numbers gives

$$\bar{Z}_N \to \mu_\theta \qquad \text{a.s.}$$

Because $\mu_\theta > 0$ and $g(x) = \sqrt{x}$ is continuous on $[0, \infty)$, the continuous mapping theorem yields

$$\widehat{L}_{\mathrm{reg}}(\theta) = g(\bar{Z}_N) \to g(\mu_\theta) = L_{\mathrm{reg}}(\theta) \qquad \text{a.s.}$$

**Root-$N$ convergence rate.** By the central limit theorem,

$$\sqrt{N}\,(\bar{Z}_N - \mu_\theta) \rightsquigarrow \mathcal{N}(0, \sigma_\theta^2).$$

Since $g$ is differentiable at $\mu_\theta > 0$ with

$$g'(\mu_\theta) = \frac{1}{2\sqrt{\mu_\theta}},$$

the delta method gives

$$\sqrt{N}\big(g(\bar{Z}_N) - g(\mu_\theta)\big) \rightsquigarrow \mathcal{N}\left(0, \big(g'(\mu_\theta)\big)^2 \sigma_\theta^2\right) = \mathcal{N}\left(0, \frac{\sigma_\theta^2}{4\mu_\theta}\right).$$

Equivalently,

$$\sqrt{N}\big(\widehat{L}_{\mathrm{reg}}(\theta) - L_{\mathrm{reg}}(\theta)\big) \rightsquigarrow \mathcal{N}\left(0, \frac{\sigma_\theta^2}{4\mu_\theta}\right).$$

Hence

$$\widehat{L}_{\mathrm{reg}}(\theta) - L_{\mathrm{reg}}(\theta) = O_p(N^{-1/2}).$$

$\square$

## C.2. Complexity of the score-corrected sampling procedure in data dimension

Our score-corrected sampling procedure has the same order of complexity with respect to data dimensionality as the unadjusted sampler. For score-based sampling methods, drawing samples with a length-$T$ trajectory costs $O(TC_0)$ for the unadjusted sampler and $O(T(C_0 + C_{\mathrm{corr}}))$ for the adjusted sampler, where $C_0$ is the cost of evaluating the base score $\nabla_x \log p_X^0(x)$ and $C_{\mathrm{corr}}$ is the cost of the correction term in Equations (11) and (26). Both $C_0$ and $C_{\mathrm{corr}}$ are model-dependent quantities. We discuss the SGLD-based Gaussian setting and the diffusion setting separately. Below, $d_X := \dim(X)$ and $d_Y := \dim(Y)$. Following the problem setup, we assume that $d_Y = O(d_X)$.

- **Gaussian experiments (SGLD sampler):** For a closed-form Gaussian score, $C_0 = O(d_X^2)$ with dense covariance, and $O(d_X)$ under independence / diagonal covariance. For $C_{\mathrm{corr}}$, if the expectation in Equation (11) is estimated with $M$ Monte Carlo samples from $p_{Y|X}$, then $C_{\mathrm{corr}} = MC_{Y|X}$, where $C_{Y|X}$ is the cost of drawing one sample from $p_{Y|X}$ and computing $\nabla_x \log p_{Y|X}(y \mid x)$. For example, for a linear-Gaussian conditional $p_{Y|X}(\cdot \mid x) = \mathcal{N}(Ax + b, \Gamma)$, we have $\nabla_x \log p_{Y|X}(y \mid x) = A^\top \Gamma^{-1}(y - Ax - b)$. After precomputing $\Gamma^{-1}$, one such evaluation costs $O(d_X d_Y)$ for dense $A$, and just $O(d_X)$ for scalar $Y$. Thus, the overall time complexity of the adjusted sampler is $O(M \cdot TC_0)$, the same order in $d_X$ as the unadjusted sampler.

- **Diffusion models (reverse diffusion sampler):** Here the same decomposition applies, but the interpretation of a length-$T$ trajectory differs from SGLD. A length-$T$ SGLD trajectory generates $T$ correlated samples along the Markov chain, whereas a length-$T$ reverse diffusion trajectory produces one generated sample. That is, drawing one diffusion sample costs $O(TC_0)$ for the unadjusted sampler, and $O(T(C_0 + C_{\mathrm{corr}}))$ for the adjusted one, since the correction score is evaluated once at each reverse-time step.

In our image experiments, $C_0$ is the cost of one score-network evaluation on an image tensor of dimension $d_X$, while $C_{\mathrm{corr}}$ is the cost of one evaluation of the correction score per Equation (26). In our implementation, $C_{\mathrm{corr}}$ is dominated by one forward pass through $p_{Y|X(t)}(y \mid x_t)$ and one backward pass to obtain the gradient with respect to $x_t$. Since $p_{Y|X(t)}(y \mid x_t)$ is implemented as a smaller two-layer U-Net encoder than the U-Net of diffusion model score-networks (see Appendix Appendix D.2.2), both are of the same order for fixed $d_Y$. Further, the dependence on $d_Y$ is negligible in our image tasks because $|Y|$ is small (4 for CelebA and 10 for CIFAR-10). Therefore, the overall time complexity is $O(T(C_0 + C_{\mathrm{corr}})) = O(2 \cdot TC_0)$ in our implementation. Again, the adjusted sampler has the same order in $d_X$ as the unadjusted sampler.

## C.3. Unbiased Gradient Estimator for $L_2(\theta)$ in Equation (8)

$$
\begin{aligned}
\nabla_\theta L_2(\theta)|_{\theta=\theta^{(t)}} &= \nabla_\theta \log \int_{\mathcal{X}} p_X^0(x) \cdot \exp\left(\int_{\mathcal{Y}} \phi_\theta(y) p_{Y|X}(y|x)\, dy\right) dx \Big|_{\theta=\theta^{(t)}} \\
&= \frac{\nabla_\theta \int_{\mathcal{X}} p_X^0(x) \cdot \exp\left(\int_{\mathcal{Y}} \phi_\theta(y) p_{Y|X}(y|x)\, dy\right) dx \Big|_{\theta=\theta^{(t)}}}{\int_{\mathcal{X}} p_X^0(u) \cdot \exp\left(\int_{\mathcal{Y}} \phi_{\theta^{(t)}}(v) p_{Y|X}(v|u)\, dv\right) du} \\
&= \int_{\mathcal{X}} \frac{p_X^0(x) \cdot \exp\left(\int_{\mathcal{Y}} \phi_{\theta^{(t)}}(y) p_{Y|X}(y|x)\, dy\right)}{\int_{\mathcal{X}} p_X^0(u) \cdot \exp\left(\int_{\mathcal{Y}} \phi_{\theta^{(t)}}(v) p_{Y|X}(v|u)\, dv\right) du} \nabla_{\theta^{(t)}}\left(\int_{\mathcal{Y}} \phi_{\theta^{(t)}}(y) p_{Y|X}(y|x)\, dy\right) dx \\
&= \int_{\mathcal{X}} p_X(x; \theta^{(t)}) \left(\int_{\mathcal{Y}} \nabla_{\theta^{(t)}} \phi_{\theta^{(t)}}(y) p_{Y|X}(y|x)\, dy\right) dx \\
&= \int_{\mathcal{X}} \int_{\mathcal{Y}} p_X(x; \theta^{(t)}) p_{Y|X}(y|x) \nabla_{\theta^{(t)}} \phi_{\theta^{(t)}}(y)\, dy\, dx
\end{aligned}
\tag{14}
$$

# D. Experimental Details

## D.1. Additional Experimental Results

In this section, we perform two sensitivity analyses and present additional experimental results for Section 4.1, Section 4.2 and Section 4.3.

D.1.1. IMPACT OF $M$ ON VARIANCE OF $\nabla_\theta L_2(\theta)$ ESTIMATOR

We investigate the effect of $M$, the number of conditional samples $\{y_{ij}\}_{j=1}^{M} \sim p_{Y|X}(\cdot \mid x_i)$ drawn for each $x_i$, in the Monte Carlo estimator of $\widehat{\nabla_\theta L_2}(\theta)$ in Equation (9). For a fixed computational budget, increasing $M$ reduces the number of distinct $x_i$ samples that can be drawn from $p_X(\cdot; \theta)$. In our setting, this tradeoff is usually undesirable: $X$ is typically higher-dimensional and harder to explore than $Y$, so allocating computation to more samples from $p_X(\cdot; \theta)$ is more important for stable training. Thus, we use $M = 1$ as an efficient default in our experiments. This choice is not intended to be universal; when $p_{Y|X}$ is highly stochastic, drawing multiple conditional samples per $x_i$ can reduce gradient variance. Since $p_{Y|X}$ is specified by the practitioner, its stochasticity can be assessed directly, and $M$ can be increased when needed.

To evaluate this tradeoff, we repeat the density-chasm synthetic experiment from Section 4.1 while varying the conditional noise level in the Gaussian transformation. Specifically, we consider $\sigma_{Y|X} \in \{1, 3, 9\}$ and keep the same reference distribution $p_X^0 = \mathcal{N}(-10, 2^2)$ and target solution $p_X^* = \mathcal{N}(-1, 3/4)$. We compare $M \in \{1, 100, 10000\}$. As shown in Figure 8, larger values of $M$ can reduce early-training variability and mitigate occasional spikes, especially in the higher-noise settings $\sigma_{Y|X} = 3$ and $\sigma_{Y|X} = 9$. However, increasing $M$ has little effect on the final values of $D_{\mathrm{KL}}(\widehat{p_X^*} \| p_X^0)$ or $\|\widehat{p_Y^*} - p_Y^1\|_2$.

We further test a multimodal conditional distribution,

$$p_{Y|X}(\cdot \mid x) = \frac{1}{2}\mathcal{N}(2x + 1 - m, 1^2) + \frac{1}{2}\mathcal{N}(2x + 1 + m, 1^2),$$

with mode separation $m \in \{1, 3, 5\}$. The results in Figure 9 show a similar pattern: even for large mode separation, increasing $M$ from 1 provides no substantial improvement in the final KL divergence or marginal constraint gap. Overall, these experiments support using $M = 1$ as a computationally efficient default, while leaving open the option of increasing $M$ in settings where the conditional transformation is particularly noisy.

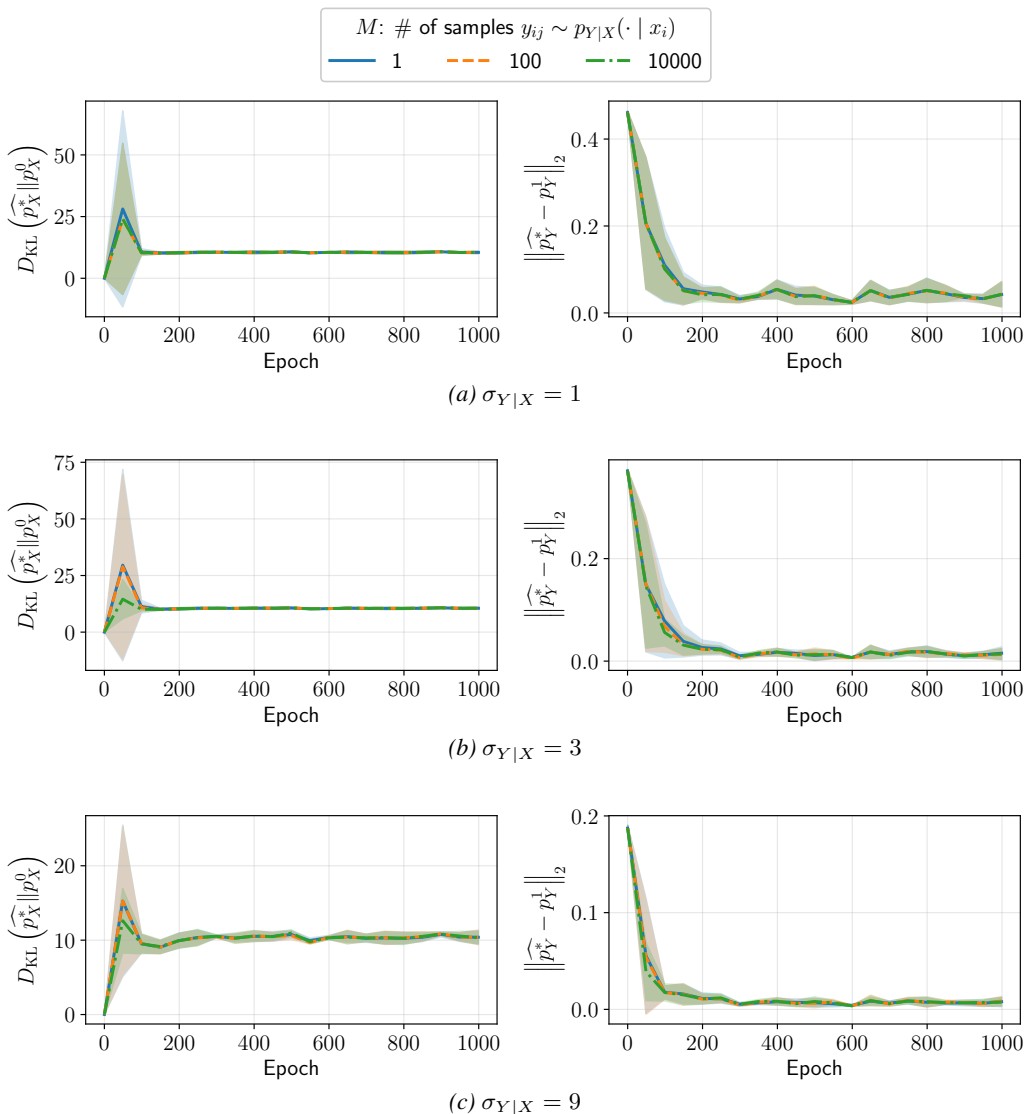

*Figure 8.* Impact of the conditional sample size $M$ under Gaussian $p_{Y|X}$ with increasing conditional noise. Larger $M$ can reduce early-training variability, especially when $\sigma_{Y|X}$ is large, but has little effect on the final KL divergence or marginal constraint gap. Curves show means over repeated runs, and shaded regions indicate one standard deviation.

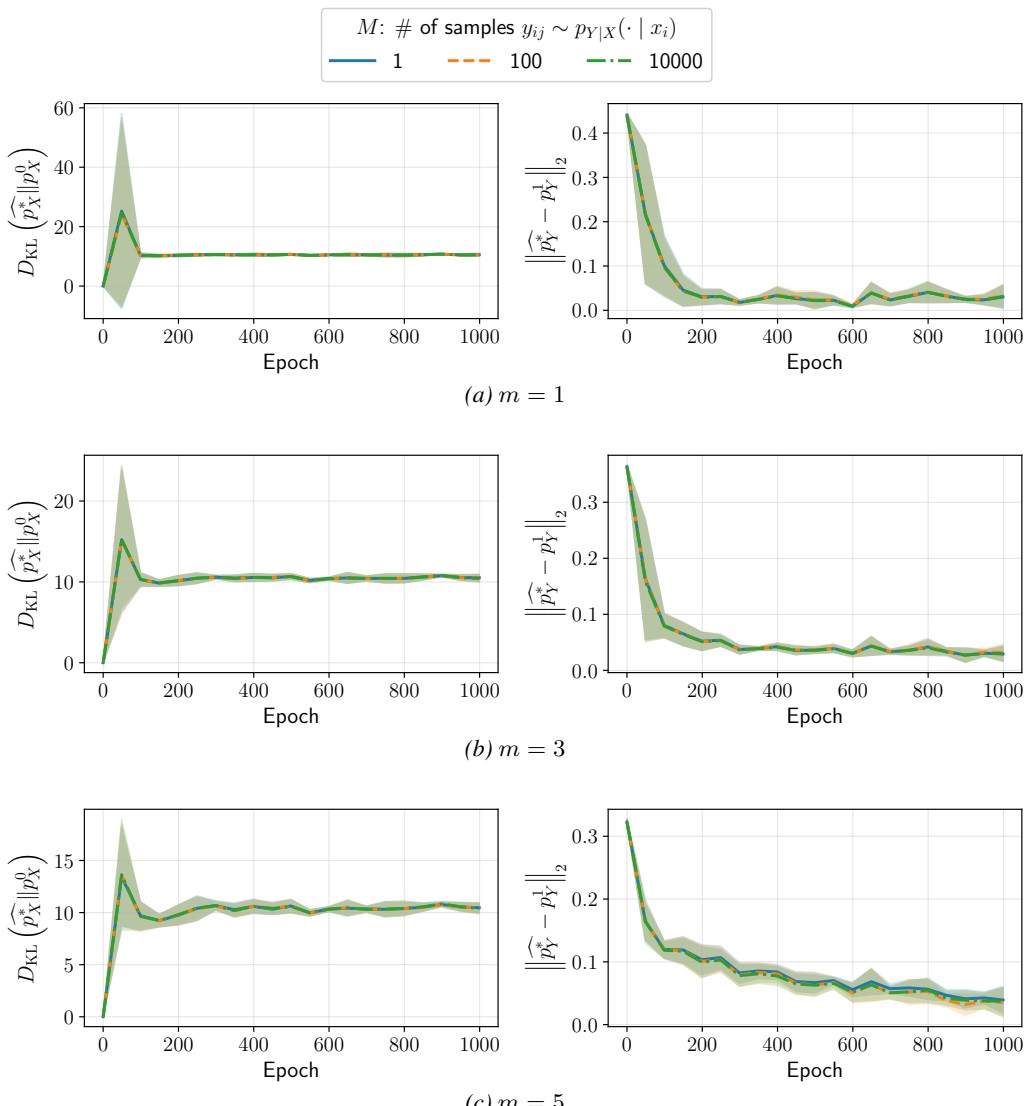

*Figure 9.* Impact of the conditional sample size $M$ under multimodal $p_{Y|X}$ with varying mode separation $m$. Increasing $M$ provides little improvement in the final KL divergence or marginal constraint gap, even when the conditional distribution is strongly multimodal. Curves show means over repeated runs, and shaded regions indicate one standard deviation.

### D.1.2. PCD SENSITIVITY ANALYSIS

As mentioned in Section 3.2, our proposed SGLD+SGD scheme essentially implements the Persistent Contrastive Divergence (PCD) framework (Tieleman, 2008). This raises a potential instability concern: if the SGD step size is too large relative to the mixing rate of the SGLD sampler, it could lead to biased ("stale") gradient estimates.

As a diagnostic, we perform a sensitivity analysis on the synthetic density-chasm experiment in Algorithm 1, with Gaussian conditional transformation and $p_X^0 = \mathcal{N}(-10, 2^2)$. We vary the number of SGLD steps per iteration (equivalently, the Monte Carlo size $N$ in Algorithm 1) while controlling the total number of samples drawn from $p_X(\cdot; \theta)$ during training. The SGLD discretization step size is fixed at $\alpha = 0.1$, as in all SGLD-based experiments.

The results show an interaction between Langevin mixing and the learning rate: as is shown in Figure 10, with a large learning rate ($10^{-3}$), training is more sensitive to the number of SGLD steps, whereas with a smaller learning rate ($10^{-4}$), the method remains stable across a broad range of SGLD steps and achieves similarly low final constraint gap even with relatively short chains. In our experiments, we mitigated this issue through standard hyperparameter tuning of the SGD

learning rate (see Appendix D.2.1).

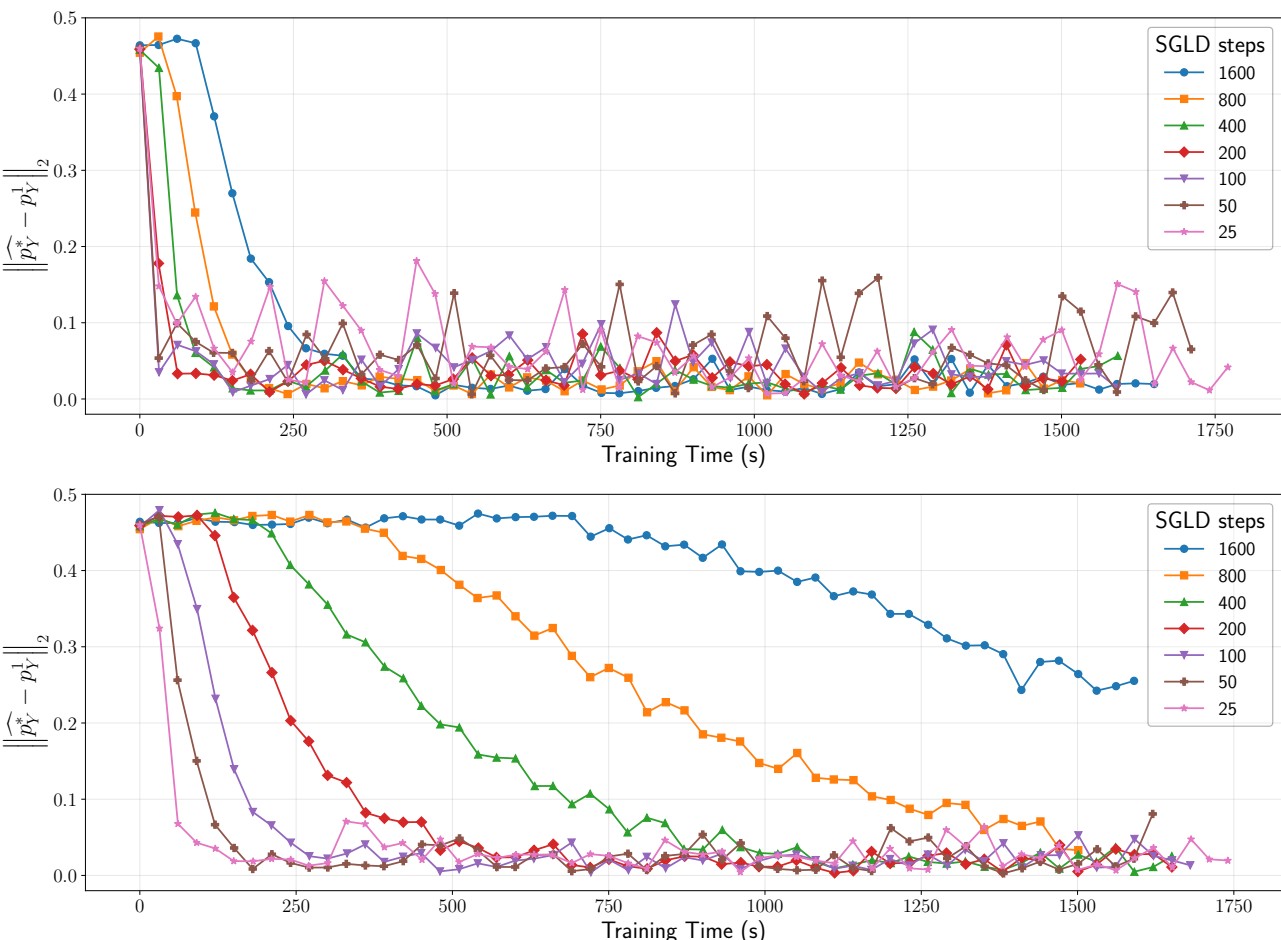

*Figure 10.* PCD sensitivity analysis on the synthetic density-chasm experiment with $p_X^0 = \mathcal{N}(-10, 2^2)$. We plot the constraint gap $\|\widehat{p_Y^*} - p_Y^1\|_2$ over training time for different numbers of SGLD steps per iteration, while controlling the total sampling budget. Top: SGD learning rate $10^{-3}$, where training is more sensitive to the number of SGLD steps. Bottom: SGD learning rate $10^{-4}$, where training is more stable across a broad range of SGLD steps.

### D.1.3. SYNTHETIC TASKS

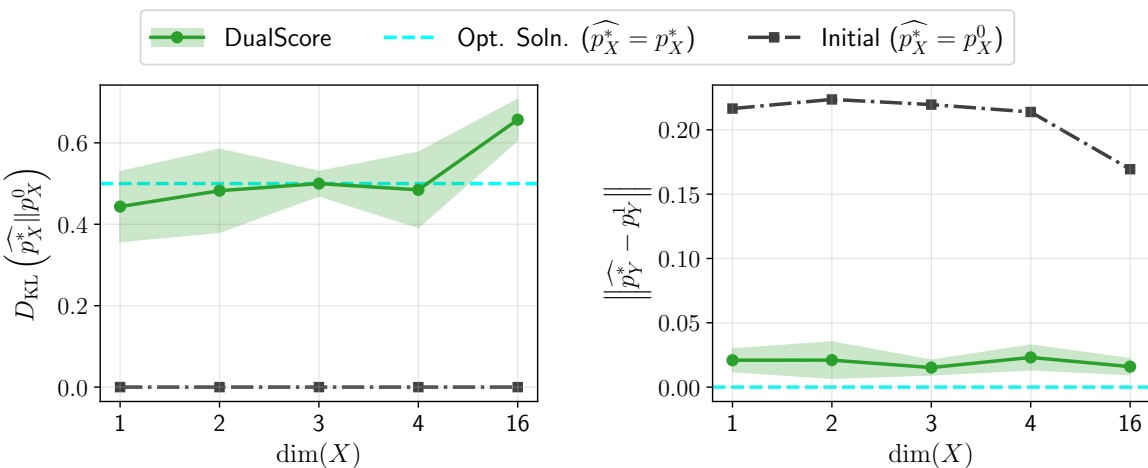

*Figure 11.* **DUALSCORE remains robust as the dimension of $X$ increases.** We consider synthetic Gaussian tasks with $X \in \mathbb{R}^d$, $p_X^0 = \mathcal{N}(\mathbf{0}, I_d)$, $p_{Y|X}(\cdot \mid \mathbf{x}) = \mathcal{N}(\mathbf{1}^\top \mathbf{x}, 1)$, and $p_Y^1 = \mathcal{N}(\sqrt{d}, d+1)$ for $d \in \{1, 2, 3, 4, 16\}$. The constrained optimum has the closed form $p_X^* = \mathcal{N}((1/\sqrt{d}, \dots, 1/\sqrt{d}), I_d)$, which satisfies $D_{\mathrm{KL}}(p_X^* \| p_X^0) = 0.5$ and $\|p_Y^* - p_Y^1\|_2 = 0$ for all $d$. Across dimensions, DUALSCORE stays close to this optimum, achieving low constraint violation while incurring nearly optimal KL divergence from the reference distribution. Lines denote means over independent runs and shaded bands denote one standard deviation.

### D.1.4. REGULARIZED NPMLE

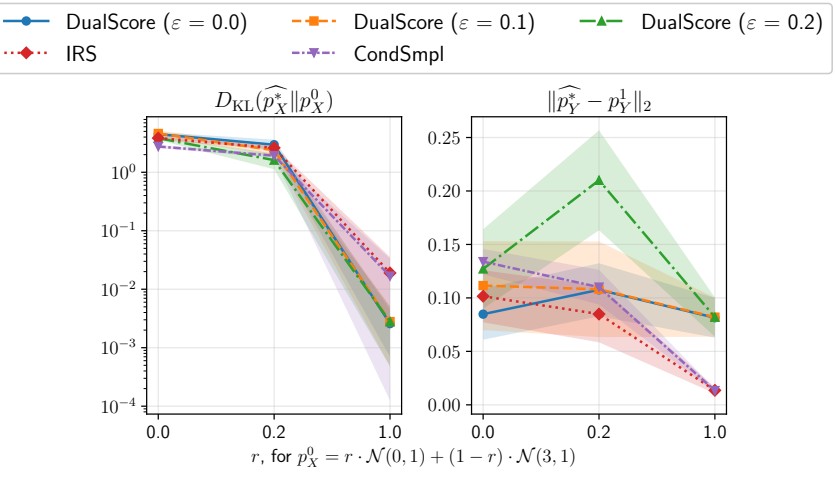

*(a)* Number of samples for $p_Y^1$ KDE = 100.

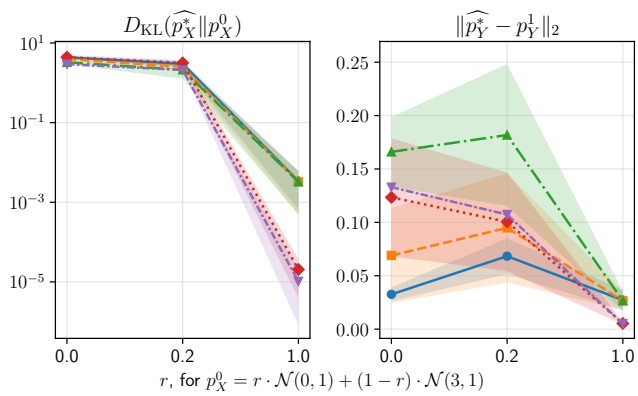

*(b)* Number of samples for $p_Y^1$ KDE = 100,000.

*Figure 12.* NPMLE results of DUALSCORE for dataset sizes 100 (top) and 100,000 (bottom) with solutions given by IRS and CONDSMPL, two baselines derived for deterministic constraints. Increasing $r$ moves the reference $p_X^0$ closer to the true latent distribution $p_X^T$, thereby reducing the density chasm between $p_X^0$ and $p_Y^1$. IRS and CondSmpl can be competitive when the transformation is nearly deterministic and the reference density is already close to the true data-generating distribution, especially at $r = 1$. DUALSCORE is most beneficial in regimes with a larger density chasm, where sampling from the corrected density cannot be reliably approximated by reweighting or conditional sampling. Lines show means over 5 runs, and shaded ribbons indicate one standard deviation.

### D.1.5. IMAGE TASKS

| Dataset | $\widehat{p_X^*}$ | FID $\left(\widehat{p_X^*}, p_X^0\right)$ (as noise-level $\rho$ increases) | | | | $\left\|\widehat{p_Y^*} - p_Y^1\right\|_2$ (as noise-level $\rho$ increases) | | | |
|---|---|---|---|---|---|---|---|---|---|
| | | $\rho = 0$ | $\rho = 0.25$ | $\rho = 0.5$ | $\rho = 0.75$ | $\rho = 0$ | $\rho = 0.25$ | $\rho = 0.5$ | $\rho = 0.75$ |
| | | | | | Balanced | | | | |
| CelebA | $p_X^0$ | 0.457 | - | - | - | 0.226 | 0.168 | 0.113 | 0.056 |
| | IRS | 1.399 | 0.896 | 0.706 | 0.668 | 0.017 | 0.088 | 0.090 | 0.054 |
| | CONDSMPL | 1.415 | 0.925 | 0.719 | 0.686 | 0.019 | 0.088 | 0.091 | 0.054 |
| | DUALSCORE | 1.159 | 1.209 | 1.049 | 1.089 | **0.011** | **0.010** | **0.009** | **0.003** |
| CIFAR-10 | $p_X^0$ | 0.947 | - | - | - | 0.158 | 0.117 | 0.079 | 0.039 |
| (LT/5%) | IRS | 4.537 | 2.602 | 1.658 | 1.418 | 0.009 | 0.053 | 0.058 | 0.037 |
| | CONDSMPL | 4.665 | 2.601 | 1.713 | 1.440 | **0.007** | 0.053 | 0.057 | 0.037 |
| | DUALSCORE | 3.936 | 4.347 | 3.957 | 4.023 | 0.014 | **0.013** | **0.007** | **0.004** |
| CIFAR-10 | $p_X^0$ | 1.016 | - | - | - | 0.184 | 0.141 | 0.091 | 0.046 |
| (LT/10%) | IRS | 4.914 | 2.610 | 1.809 | 1.507 | **0.012** | 0.065 | 0.068 | 0.044 |
| | CONDSMPL | 4.946 | 2.687 | 1.836 | 1.523 | **0.012** | 0.065 | 0.068 | 0.043 |
| | DUALSCORE | 4.086 | 4.870 | 4.296 | 4.719 | 0.018 | **0.013** | **0.008** | **0.003** |
| CIFAR-10 | $p_X^0$ | 1.005 | - | - | - | 0.205 | 0.154 | 0.103 | 0.051 |
| (LT/25%) | IRS | 5.456 | 2.786 | 1.756 | 1.523 | **0.014** | 0.075 | 0.079 | 0.048 |
| | CONDSMPL | 5.645 | 2.842 | 1.863 | 1.556 | **0.014** | 0.073 | 0.078 | 0.047 |
| | DUALSCORE | 5.318 | 5.316 | 5.302 | 5.442 | 0.015 | **0.012** | **0.008** | **0.005** |
| CIFAR-10 | $p_X^0$ | 0.978 | - | - | - | 0.197 | 0.147 | 0.098 | 0.049 |
| (LT/50%) | IRS | 5.370 | 2.730 | 1.756 | 1.488 | **0.010** | 0.070 | 0.073 | 0.045 |
| | CONDSMPL | 5.371 | 2.748 | 1.757 | 1.492 | **0.010** | 0.068 | 0.075 | 0.046 |
| | DUALSCORE | 5.220 | 5.228 | 5.434 | 5.312 | 0.019 | **0.014** | **0.011** | **0.005** |
| | | | | | Reversed | | | | |
| CelebA | $p_X^0$ | 0.457 | - | - | - | 0.444 | 0.388 | 0.332 | 0.277 |
| | IRS | 3.395 | 1.611 | 0.975 | 0.711 | 0.040 | 0.207 | 0.269 | 0.264 |
| | CONDSMPL | 3.416 | 1.738 | 1.078 | 0.781 | 0.040 | 0.207 | 0.266 | 0.264 |
| | DUALSCORE | 3.094 | 5.100 | 61.213 | 127.591 | **0.038** | **0.059** | **0.031** | **0.097** |
| CIFAR-10 | $p_X^0$ | 0.947 | - | - | - | 0.283 | 0.249 | 0.214 | 0.185 |
| (LT/5%) | IRS | 12.819 | 5.704 | 2.984 | 1.733 | **0.019** | 0.127 | 0.168 | 0.176 |
| | CONDSMPL | 12.864 | 6.136 | 3.051 | 1.882 | 0.021 | 0.125 | 0.171 | 0.176 |
| | DUALSCORE | 12.713 | 17.850 | 40.707 | 113.409 | 0.046 | **0.040** | **0.038** | **0.041** |
| CIFAR-10 | $p_X^0$ | 1.016 | - | - | - | 0.330 | 0.291 | 0.250 | 0.216 |
| (LT/10%) | IRS | 10.750 | 4.955 | 2.489 | 1.686 | 0.033 | 0.156 | 0.204 | 0.206 |
| | CONDSMPL | 11.003 | 5.062 | 2.875 | 1.906 | **0.031** | 0.157 | 0.201 | 0.206 |
| | DUALSCORE | 9.567 | 13.241 | 48.051 | 83.789 | 0.059 | **0.070** | **0.048** | **0.102** |
| CIFAR-10 | $p_X^0$ | 1.005 | - | - | - | 0.368 | 0.321 | 0.278 | 0.240 |
| (LT/25%) | IRS | 11.792 | 5.056 | 2.565 | 1.669 | 0.037 | 0.175 | 0.227 | 0.230 |
| | CONDSMPL | 11.956 | 5.345 | 2.847 | 1.905 | **0.032** | 0.175 | 0.227 | 0.231 |
| | DUALSCORE | 11.813 | 16.525 | 29.165 | 55.331 | 0.064 | **0.048** | **0.038** | **0.091** |
| CIFAR-10 | $p_X^0$ | 0.978 | - | - | - | 0.356 | 0.313 | 0.270 | 0.231 |
| (LT/50%) | IRS | 11.922 | 4.827 | 2.432 | 1.648 | 0.022 | 0.169 | 0.220 | 0.221 |
| | CONDSMPL | 12.051 | 5.173 | 2.770 | 1.898 | **0.020** | 0.169 | 0.219 | 0.221 |
| | DUALSCORE | 12.803 | 20.013 | 102.797 | 82.481 | 0.067 | **0.066** | **0.036** | **0.070** |

*Table 2.* Complete results for the image tasks.

| Dataset | $\widehat{p_X^*}$ | FID $\left(\widehat{p_X^*}, p_X^0\right)$ (as noise-level $\rho$ increases) | | | | $\left\|\widehat{p_Y^*} - p_Y^1\right\|_2$ (as noise-level $\rho$ increases) | | | |
|---|---|---|---|---|---|---|---|---|---|
| | | $\rho = 0$ | $\rho = 0.25$ | $\rho = 0.5$ | $\rho = 0.75$ | $\rho = 0$ | $\rho = 0.25$ | $\rho = 0.5$ | $\rho = 0.75$ |
| | | | | | Balanced | | | | |
| CelebA | $p_X^0$ | 0.460±0.006 | - | - | - | 0.223±0.002 | 0.167±0.001 | 0.112±0.001 | 0.056±0.000 |
| | IRS | 1.388±0.033 | 0.872±0.015 | 0.719±0.010 | 0.680±0.016 | 0.019±0.001 | 0.088±0.001 | 0.089±0.001 | 0.054±0.001 |
| | CONDSMPL | 1.411±0.014 | 0.909±0.013 | 0.731±0.014 | 0.691±0.008 | 0.017±0.001 | 0.086±0.002 | 0.089±0.001 | 0.053±0.001 |
| | DUALSCORE | 1.160±0.082 | 1.130±0.074 | 1.184±0.117 | 1.149±0.058 | **0.016**±0.006 | **0.011**±0.003 | **0.010**±0.003 | **0.003**±0.001 |
| CIFAR-10 (LT/5%) | $p_X^0$ | 0.966±0.015 | - | - | - | 0.156±0.002 | 0.117±0.001 | 0.078±0.001 | 0.039±0.001 |
| | IRS | 4.680±0.077 | 2.493±0.070 | 1.697±0.022 | 1.467±0.029 | 0.008±0.001 | 0.053±0.001 | 0.058±0.001 | 0.036±0.001 |
| | CONDSMPL | 4.730±0.066 | 2.580±0.020 | 1.727±0.021 | 1.483±0.025 | **0.008**±0.000 | 0.052±0.001 | 0.057±0.001 | 0.036±0.001 |
| | DUALSCORE | 4.087±0.278 | 4.130±0.194 | 4.128±0.152 | 4.135±0.134 | 0.019±0.004 | **0.015**±0.002 | **0.010**±0.001 | **0.005**±0.001 |
| CIFAR-10 (LT/10%) | $p_X^0$ | 1.010±0.005 | - | - | - | 0.185±0.002 | 0.139±0.002 | 0.092±0.001 | 0.046±0.001 |
| | IRS | 4.872±0.109 | 2.603±0.019 | 1.792±0.022 | 1.528±0.020 | 0.013±0.001 | 0.066±0.001 | 0.069±0.001 | 0.043±0.000 |
| | CONDSMPL | 4.985±0.028 | 2.692±0.061 | 1.811±0.042 | 1.536±0.014 | **0.012**±0.001 | 0.065±0.001 | 0.069±0.001 | 0.043±0.000 |
| | DUALSCORE | 4.520±0.252 | 4.685±0.296 | 4.341±0.262 | 4.636±0.297 | 0.020±0.003 | **0.016**±0.001 | **0.011**±0.001 | **0.006**±0.001 |
| CIFAR-10 (LT/25%) | $p_X^0$ | 1.000±0.006 | - | - | - | 0.206±0.003 | 0.155±0.002 | 0.103±0.001 | 0.103±0.001 |
| | IRS | 5.530±0.107 | 2.788±0.047 | 1.798±0.048 | 1.798±0.048 | **0.014**±0.001 | 0.074±0.001 | 0.078±0.002 | 0.078±0.002 |
| | CONDSMPL | 5.621±0.052 | 2.835±0.023 | 1.815±0.040 | 1.815±0.040 | **0.014**±0.001 | 0.074±0.001 | 0.078±0.001 | 0.078±0.001 |
| | DUALSCORE | 5.364±0.156 | 5.333±0.129 | 5.385±0.155 | 5.422±0.151 | 0.020±0.002 | **0.015**±0.001 | **0.010**±0.001 | **0.005**±0.001 |
| CIFAR-10 (LT/50%) | $p_X^0$ | 0.985±0.006 | - | - | - | 0.199±0.002 | 0.149±0.001 | 0.099±0.001 | 0.050±0.000 |
| | IRS | 5.193±0.091 | 2.664±0.042 | 1.735±0.015 | 1.489±0.022 | 0.011±0.001 | 0.070±0.000 | 0.075±0.001 | 0.046±0.001 |
| | CONDSMPL | 5.322±0.027 | 2.731±0.036 | 1.764±0.015 | 1.499±0.011 | **0.010**±0.001 | 0.070±0.001 | 0.075±0.001 | 0.046±0.001 |
| | DUALSCORE | 5.080±0.151 | 5.104±0.134 | 5.049±0.168 | 5.085±0.270 | 0.019±0.002 | **0.015**±0.001 | **0.010**±0.001 | **0.005**±0.000 |
| | | | | | Reversed | | | | |
| CelebA | $p_X^0$ | 0.460±0.006 | - | - | - | 0.442±0.002 | 0.386±0.001 | 0.331±0.001 | 0.277±0.000 |
| | IRS | 3.365±0.035 | 1.591±0.023 | 0.947±0.018 | 0.722±0.018 | 0.041±0.003 | 0.207±0.001 | 0.267±0.001 | 0.264±0.001 |
| | CONDSMPL | 3.412±0.029 | 1.724±0.023 | 1.036±0.023 | 0.801±0.016 | **0.038**±0.001 | 0.204±0.002 | 0.267±0.001 | 0.264±0.001 |
| | Ours | 3.327±0.211 | 4.858±0.731 | 61.880±14.014 | 130.793±18.425 | 0.050±0.007 | **0.060**±0.004 | **0.035**±0.003 | **0.100**±0.002 |
| CIFAR-10 (LT/5%) | $p_X^0$ | 0.966±0.015 | - | - | - | 0.282±0.002 | 0.248±0.001 | 0.215±0.001 | 0.185±0.000 |
| | IRS | 12.587±0.199 | 5.815±0.072 | 2.872±0.064 | 1.721±0.041 | 0.021±0.001 | 0.127±0.000 | 0.170±0.001 | 0.176±0.000 |
| | CONDSMPL | 12.746±0.063 | 6.149±0.068 | 3.056±0.004 | 1.876±0.049 | **0.021**±0.001 | 0.125±0.001 | 0.171±0.000 | 0.176±0.001 |
| | DUALSCORE | 13.256±1.159 | 19.113±2.467 | 40.004±5.831 | 114.085±2.095 | 0.054±0.003 | **0.046**±0.002 | **0.041**±0.003 | **0.042**±0.000 |
| CIFAR-10 (LT/10%) | $p_X^0$ | 1.010±0.005 | - | - | - | 0.331±0.002 | 0.290±0.001 | 0.251±0.001 | 0.215±0.000 |
| | IRS | 11.018±0.195 | 4.898±0.103 | 2.525±0.076 | 1.706±0.024 | **0.031**±0.002 | 0.157±0.001 | 0.205±0.001 | 0.206±0.000 |
| | CONDSMPL | 11.167±0.172 | 5.207±0.129 | 2.824±0.076 | 1.919±0.035 | 0.032±0.001 | 0.155±0.002 | 0.204±0.001 | 0.206±0.001 |
| | DUALSCORE | 8.891±0.715 | 13.711±2.114 | 54.909±5.906 | 102.469±13.443 | 0.068±0.004 | **0.077**±0.007 | **0.055**±0.006 | **0.104**±0.002 |
| CIFAR-10 (LT/25%) | $p_X^0$ | 1.000±0.006 | - | - | - | 0.369±0.002 | 0.323±0.002 | 0.280±0.001 | 0.240±0.000 |
| | IRS | 11.950±0.157 | 4.984±0.085 | 2.507±0.035 | 1.651±0.018 | 0.034±0.001 | 0.178±0.003 | 0.229±0.001 | 0.230±0.001 |
| | CONDSMPL | 12.119±0.140 | 5.346±0.083 | 2.829±0.038 | 1.944±0.028 | **0.033**±0.002 | 0.177±0.002 | 0.229±0.001 | 0.230±0.001 |
| | DUALSCORE | 10.986±0.728 | 16.524±0.428 | 90.805±36.366 | 52.424±4.177 | 0.063±0.004 | **0.050**±0.001 | **0.045**±0.004 | **0.093**±0.002 |
| CIFAR-10 (LT/50%) | $p_X^0$ | 0.984±0.006 | - | - | - | 0.357±0.002 | 0.313±0.001 | 0.270±0.001 | 0.231±0.000 |
| | IRS | 11.927±0.066 | 4.828±0.109 | 2.359±0.062 | 1.610±0.020 | 0.021±0.001 | 0.169±0.002 | 0.222±0.001 | 0.221±0.000 |
| | CONDSMPL | 12.064±0.095 | 5.188±0.105 | 2.740±0.037 | 1.873±0.026 | **0.020**±0.001 | 0.168±0.001 | 0.220±0.001 | 0.221±0.000 |
| | DUALSCORE | 13.420±0.612 | 20.150±0.201 | 83.117±27.097 | 81.829±1.569 | 0.073±0.004 | **0.065**±0.002 | **0.044**±0.006 | **0.070**±0.000 |

*Table 3.* Complete results for the image tasks averaged over five runs (± standard deviation).

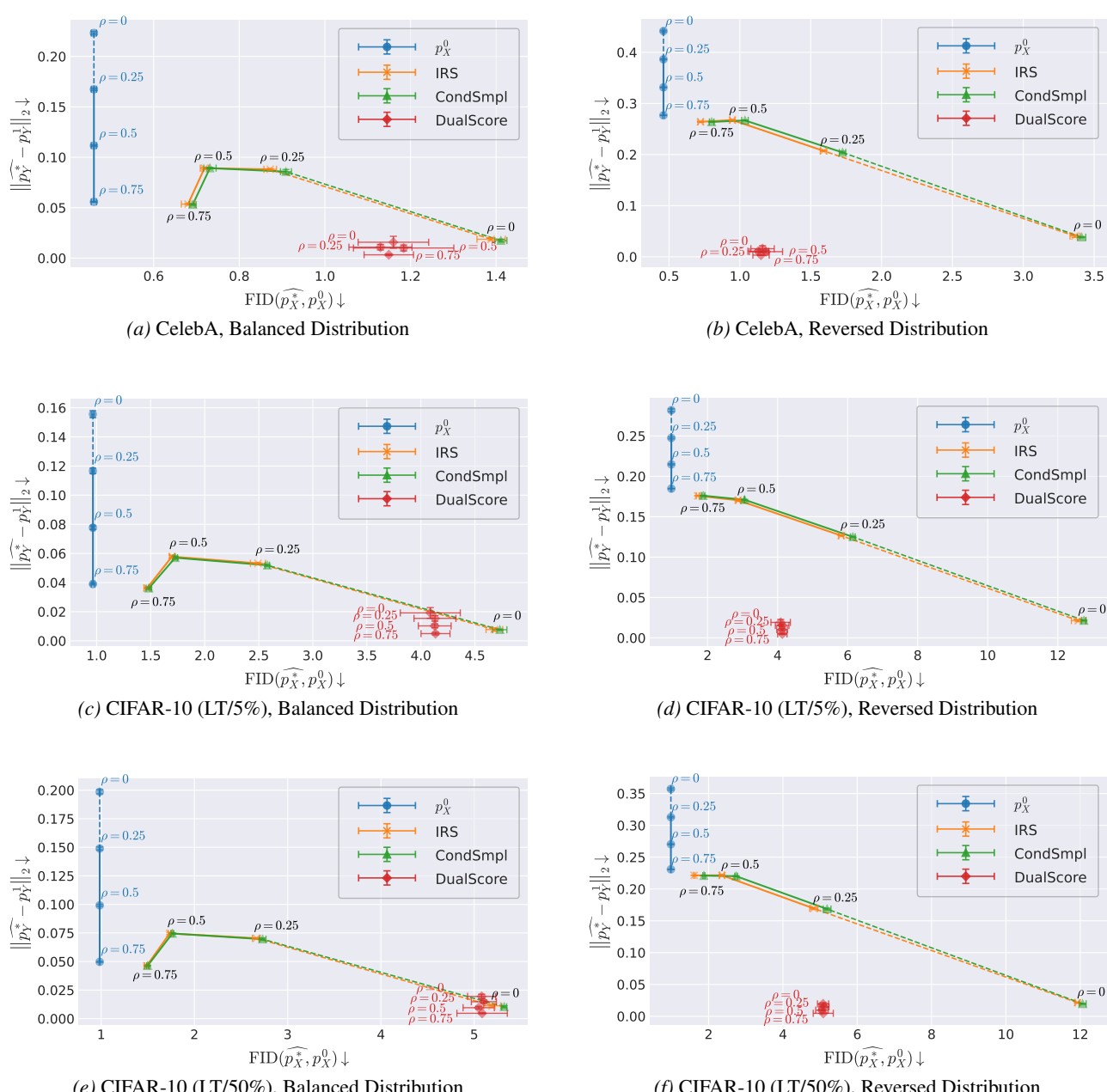

*Figure 13.* **Our method remains robust to noisy transformations across tasks and datasets, while baselines designed for deterministic constraints degrade as the transformation becomes noisier.** We visualize the FID–L2 tradeoff under noisy classifier transformations for the five-run results reported in Table 3. Each panel shows the tradeoff between fidelity to the original generator, measured by $\mathrm{FID}(\widehat{p_X^*}, p_X^0)$, and marginal constraint violation, measured by $\|\widehat{p_Y^*} - p_Y^1\|_2$, under a different $(p_X^0, p_Y^1)$ setup. Error bars denote one standard deviation across 5 runs with different seeds. When $\rho = 0$, DUALSCORE achieves comparable L2 discrepancy to IRS and CONDSMPL with lower FID; when $\rho > 0$, it substantially outperforms both baselines in constraint violation, with the larger FID reflecting the greater distributional shift required by the constrained solution. **This highlights the advantage of directly modeling probabilistic constraints when the classifier transformation is noisy.**

**CelebA, Balanced Distribution.**

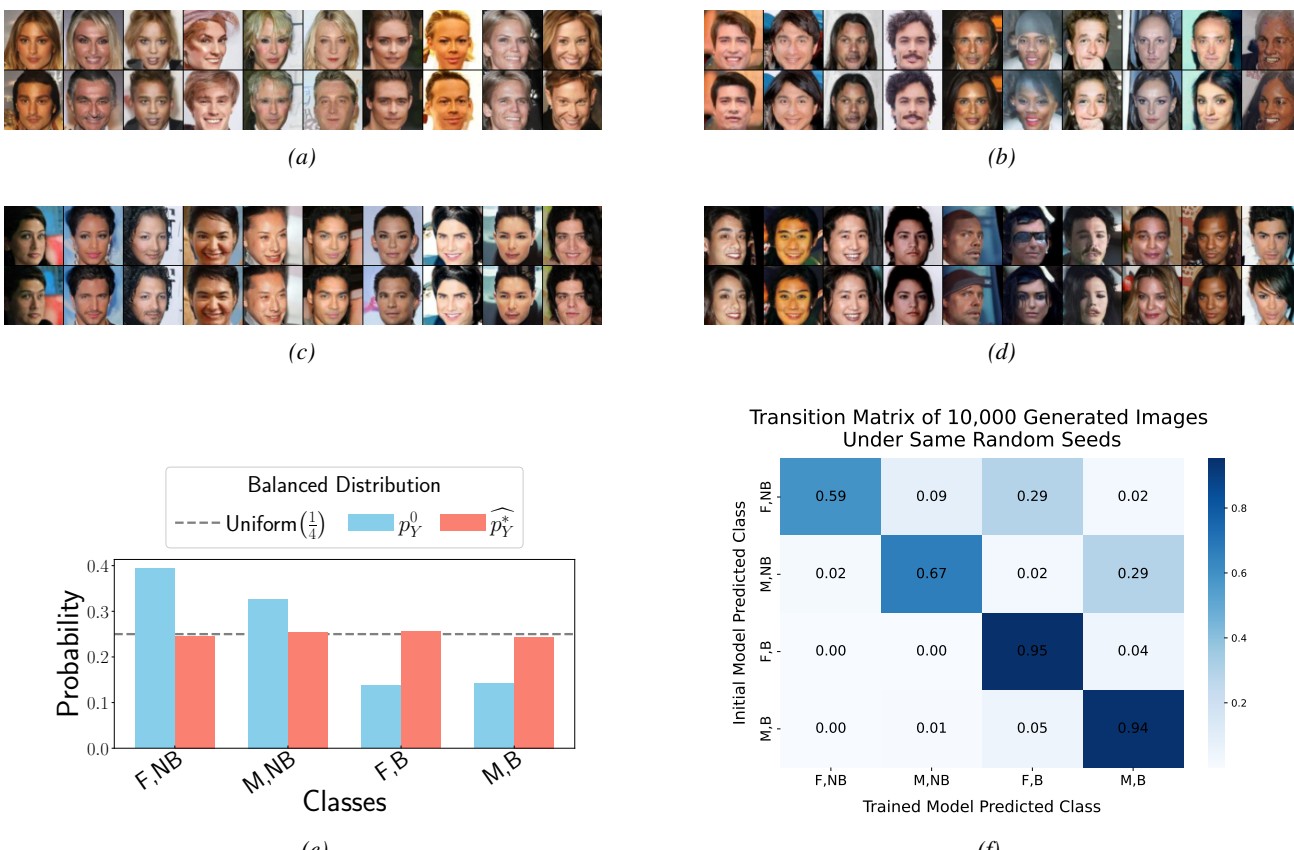

*Figure 14.* Our method effectively balances the class distribution of CelebA with respect to the target attributes (gender and hair color). Panels (a–d) show samples generated with identical random seeds before and after correction, illustrating instance-level attribute conversion across gender and hair color. Panels (e) and (f) report the induced class distribution shift and transition dynamics, respectively. Particularly, panel (f) reveals a clear direction in distributional-level correction: probability mass is transferred from the major classes to the minor classes, while samples initially assigned to these minority classes largely remain in place.

**CIFAR-10 (LT/5%), Balanced Distribution.**

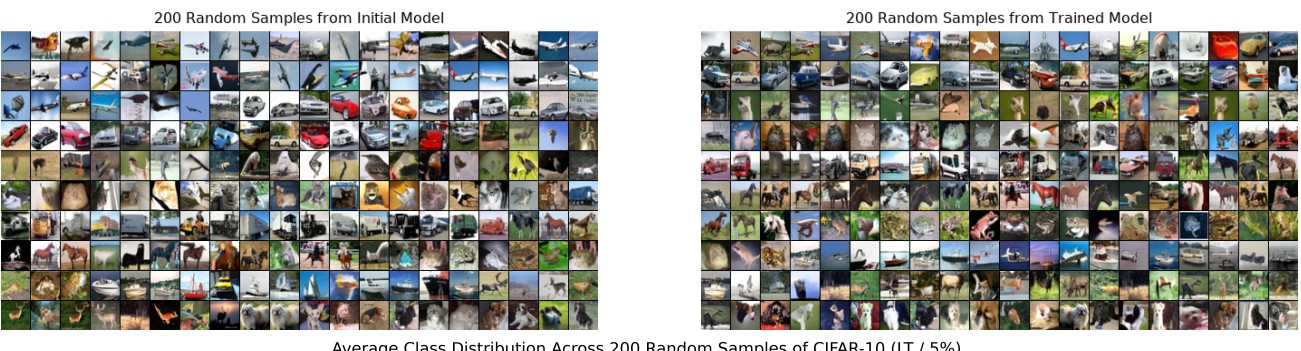

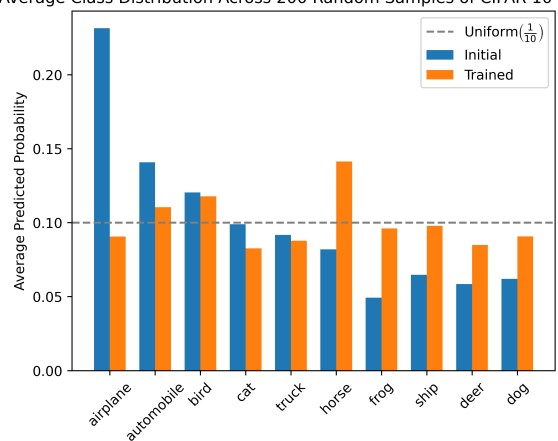

*Figure 15.* Our proposal effectively corrects the biased class distribution of CIFAR-10 (LT/5%). From 200 randomly sampled images, we observe that image quality is largely preserved while the class distribution is significantly more balanced.

**CIFAR-10 (LT/5%), Reversed Distribution.**

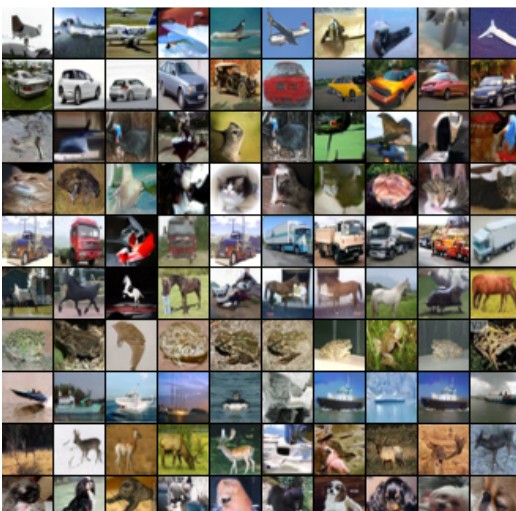
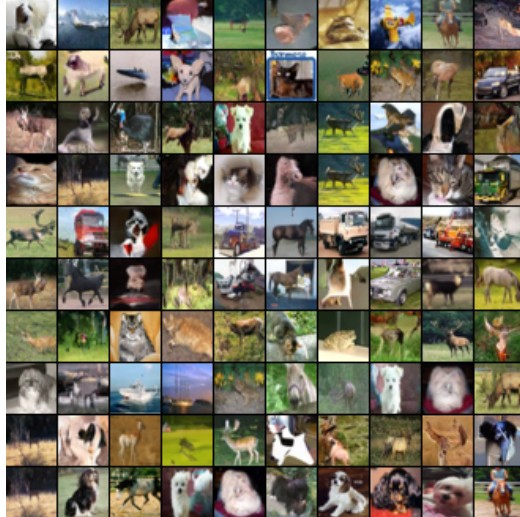

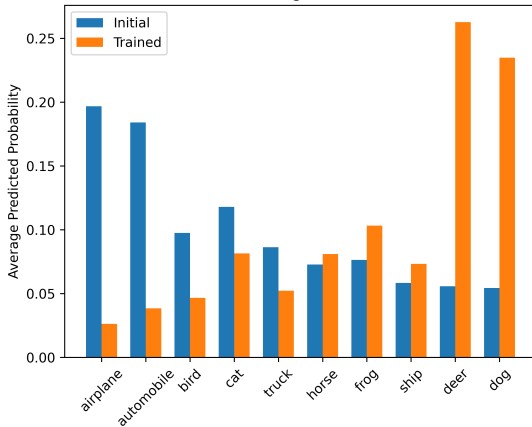

*Figure 16.* Our proposal effectively flips the class distribution of CIFAR-10 (LT/5%). Using samples generated under the same random seeds, we observe that image quality is largely preserved while the class distribution is reversed in accordance with the marginal constraint. In particular, most airplanes and automobiles are transformed into underrepresented classes, whereas rare classes such as deer and dogs tend to remain unchanged.

## D.2. Implementation details

Most experiments are implemented in PyTorch (Paszke et al., 2019). For the parametrized dual variable $\phi_\theta$, we use the multilayer perceptron (MLP) implementation from TorchRL (Bou et al., 2024)[1] and apply an additional small-weight initialization step: after standard PyTorch `nn.Linear` initialization, we multiply all linear weights and biases by 0.1. This yields near-zero initial $\phi_{\theta^{(0)}}$, which ensures $\nabla_x \log p_X(x; \theta^{(0)}) \approx \nabla_x \log p_X^0(x)$ as outlined in Section 3.

Unless otherwise specified, experiments were run on the Purdue Rosen Center for Advanced Computing (RCAC) Gilbreth cluster[2], where each training job uses a single available NVIDIA A10, A30, or A100-40GB GPUs. The main exception is the pretraining of the reference diffusion models $p_X^0$ for the image tasks: the CIFAR-10 and CelebA reference diffusion models are trained using 2 and 4 A100-40GB GPUs, respectively.

### D.2.1. SYNTHETIC AND NPMLE TASKS

For all synthetic tasks, we assume that $p_X^0(x)$, $\nabla_x \log p_X^0(x)$, $p_{Y|X}(y|x)$, and $p_Y^1(y)$ are available for exact sampling and density evaluation. For NPMLE tasks, the main difference is that $p_Y^1 = \text{KDE}(\{y_i\}_{i=1}^N)$, where $y_i \sim p_{Y|X}(\cdot|x_i)$ and $x_i \sim p_X^T$. We parameterize $\phi_\theta(\cdot) : \mathcal{Y} \to \mathbb{R}$ using MLPs with two fully connected hidden layers of widths $(32, 16)$, each followed by a ReLU activation.

**Optimal Solutions.** Specifically, we consider the problem (1) with $p_X^0(x) \coloneqq \mathcal{N}(\mu^0, (\sigma_X^0)^2)$, $p_{Y|X}(y|x) \coloneqq \mathcal{N}(2x+1, 1^2)$ and $p_Y^1(y) = \mathcal{N}(-1, \sigma_Y^{1\,2})$, under the strict constraint $\varepsilon = 0$. When $\sigma_Y^1 > 1$, the problem is feasible and the solution is given by

$$p_X^* = \mathcal{N}\left(-1, \left(\frac{\sqrt{\sigma_Y^{*\,2} - 1}}{2}\right)^2\right) \coloneqq \mathcal{N}(\mu_X^*, (\sigma_X^*)^2) \tag{15}$$

To see this, note that mean $\mu_X^*$ and variance $(\sigma_X^*)^2$ of $p_X^*$ are uniquely determined by $2 \cdot \mu_X^* + 1 = -1$ and $2^2 \cdot (\sigma_X^*)^2 + 1 = \sigma_Y^{*\,2}$. It follows from simple calculations that KL divergence is minimized when $p_X^*$ is a normal distribution.[3]

**Conditional Sampling.** We describe how $p_{X|Y}^0$ is sampled in our experiments for CONDSMPL in the synthetic and NPMLE tasks.

When both $p_Y^1$ and $p_{Y|X}$ are Gaussian, $p_{X|Y}^0$ can be calculated in closed-form for exact sampling. Specifically, let $p_X^0 = \mathcal{N}(\mu_0, \Sigma_0)$ and $p_{Y|X} = \mathcal{N}(\mathbf{a}^\top X + b, \sigma_{Y|X}^2)$, where $\mu_0, \mathbf{a} \in \mathbb{R}^d$, $\Sigma_0 \in \mathbb{R}^{d \times d}$, and $b, \sigma_{Y|X} \in \mathbb{R}$. Note that $Y$ can be equivalently represented as a linear transformation

$$Y = \tilde{\mathbf{a}}^\top \widetilde{X} + b,$$

where

$$\tilde{\mathbf{a}} \coloneqq \begin{pmatrix} \mathbf{a} \\ 1 \end{pmatrix}, \qquad \widetilde{X} \coloneqq \begin{pmatrix} X \\ \epsilon \end{pmatrix} \sim \mathcal{N}\left(\begin{pmatrix} \mu_0 \\ 0 \end{pmatrix}, \begin{pmatrix} \Sigma_0 & \mathbf{0} \\ \mathbf{0}^\top & \sigma_{Y|X}^2 \end{pmatrix}\right) \equiv \mathcal{N}(\tilde{\mu}, \tilde{\Sigma}).$$

Since this transformation is linear, $(Y, \widetilde{X})$ is jointly Gaussian. It follows that the conditional distribution of $\widetilde{X}$ given $Y$ is

$$p_{\widetilde{X}|Y}^0 \sim \mathcal{N}(\mu_{\widetilde{X}|Y}, \Sigma_{\widetilde{X}|Y}),$$

with

$$\mu_{\widetilde{X}|Y} = \tilde{\mu} + \tilde{\Sigma}\tilde{\mathbf{a}}(\tilde{\mathbf{a}}^\top \tilde{\Sigma}\tilde{\mathbf{a}})^{-1}\left(Y - (\tilde{\mathbf{a}}^\top \tilde{\mu} + b)\right),$$

$$\Sigma_{\widetilde{X}|Y} = \tilde{\Sigma} - \tilde{\Sigma}\tilde{\mathbf{a}}(\tilde{\mathbf{a}}^\top \tilde{\Sigma}\tilde{\mathbf{a}})^{-1}\tilde{\mathbf{a}}^\top \tilde{\Sigma}.$$

Finally, we sample $x \sim X|Y \equiv \mathcal{N}(\mu_{\widetilde{X}|Y,1:d}, \Sigma_{\widetilde{X}|Y,1:d})$ – the posterior distribution of $\widetilde{X}|Y$ excluding the noise $\epsilon$.

---

[1] https://github.com/pytorch/rl/blob/main/torchrl/modules/models/models.py
[2] https://www.rcac.purdue.edu/compute/gilbreth
[3] https://math.stackexchange.com/a/2354469/719714

When the Gaussian conditions are violated—for example, when $p_{Y|X}$ follows a Laplace distribution or when $p_Y^1$ is estimated by KDE—we sample from the posterior $p_{X|Y}^0$ using a Metropolis–Hastings algorithm with a simple symmetric proposal, such as $\mathcal{N}(x, 1)$. For high-dimensional or more complex settings, more expressive parameterized proposal models are likely required for efficient sampling.

**Training and model selection.** We conduct five independent runs with different random seeds for each hyperparameter setting. In particular, we search over the following key hyperparameters:

- Learning rate: $\{10^{-5}, 10^{-4}, 10^{-3}\}$.

- Monte Carlo size $N$ (in Algorithm 1): $\{16, 32, 64\}$.

During training, we evaluate each run at fixed intervals for $D_{\mathrm{KL}}(\widehat{p_X^*}\|p_X^0)$ and $\|\widehat{p_Y^*} - p_Y^1\|_2$; we refer to these as *validation* KL and *validation* $L^2$ distances. For each run, we select the best checkpoint using a *constraint-first* rule: if at least one checkpoint satisfies $\|\widehat{p_Y^*} - p_Y^1\|_2 \leq \varepsilon$, we choose, among those checkpoints, the one with the lowest validation KL. Otherwise, we choose the checkpoint with the lowest validation $L^2$ distance. We then aggregate the selected validation metrics across the five runs for each hyperparameter setting, and choose the setting with the largest number of constraint-satisfying runs. Ties are broken by the lowest average validation KL when all five runs satisfy the constraint, and by the lowest average validation $L^2$ distance otherwise.

After selecting hyperparameters and checkpoints, we re-evaluate the selected models using independent evaluation seeds and report the mean and standard deviation of the KL and $L^2$ metrics, as in Figures 1 and 3. Note that *validation* here refers only to periodic checkpoint evaluation. Unlike standard supervised-learning settings, our experiments do not use a fixed train/validation/test split, since optimization and evaluation samples are generated on the fly from the specified distributions.

**KL Divergence.** For synthetic and NPMLE tasks, we need to estimate $D_{\mathrm{KL}}\left(\widehat{p_X^*}\|p_X^0\right)$, where $p_X^0$ admits a known density function. This estimation is challenging as it requires evaluating the density ratio $\frac{\widehat{p_X^*}(x)}{p_X(x)}$. As a simple approximation, we fit a Gaussian to samples from $\widehat{p_X^*}$ using their empirical mean and covariance, and then use the closed-form formula for the KL divergence between Gaussians: for $p = \mathcal{N}(\mu_1, \Sigma_1)$ and $q = \mathcal{N}(\mu_2, \Sigma_2)$,

$$D_{\mathrm{KL}}(p\|q) = \tfrac{1}{2}\left[\log\frac{|\Sigma_2|}{|\Sigma_1|} + \mathrm{Tr}(\Sigma_2^{-1}\Sigma_1) + (\mu_1 - \mu_2)^{\top}\Sigma_2^{-1}(\mu_1 - \mu_2) - \dim(X)\right].$$

We also consider a KDE-based estimate: we first approximate $\widehat{p_{X,\mathrm{KDE}}^*}$ by samples from $\widehat{p_X^*}$, and then compute

$$D_{\mathrm{KL}}\left(\widehat{p_X^*}\|p_X^0\right) \approx \mathbb{E}_{x\sim\widehat{p_X^*}(x)}\left[\log\frac{\widehat{p_{X,\mathrm{KDE}}^*}(x)}{p_X^0(x)}\right].$$

In most settings, these two estimators agree closely. However, for certain tasks such as NPMLE with $r = 0.2$, the optimal solution is multi-modal, in which case KDE-based estimates are more appropriate. For high-dimensional cases, Gaussian approximation usually results in more robust estimates than KDE.

**L2 Distance.** To estimate $\left\|\widehat{p_Y^*} - p_Y^1\right\|_2$, we first approximate

$$\widehat{p_Y^*}(y) = \mathbb{E}_{x\sim\widehat{p_X^*}(x)}[p_{Y|X}(y|x)].$$

When $Y$ is discrete, $\left\|\widehat{p_Y^*} - p_Y^1\right\|_2$ reduces to the Euclidean distance between two finite-dimensional vectors. When $Y$ is continuous, however, there are several possible estimators; we use numerical integration, which is feasible in our experiments because $Y$ is one-dimensional.

### D.2.2. IMAGE TASKS

For image tasks, we first train diffusion models on biased data to represent $p^0_{X(t)}$, together with time-dependent classifiers $p_{Y|X(t)}$. We follow the training scheme and hyperparameter configurations in (Kim et al., 2024, Table 6). Specifically, $p^0_{X(t)}$ is implemented using EDM (Karras et al., 2022) with its second-order ODE sampler. The classifier $p_{Y|X(t)}$ is implemented as a two-layer U-Net encoder: the first layer is a fixed feature extractor with weights imported from the public ADM guided-diffusion implementation (Dhariwal & Nichol, 2021)[4], while the second layer is a shallow, trainable U-Net encoder (Ronneberger et al., 2015) with a softmax output whose dimension equals the number of target classes. The classifier is trained on the full training sets of CIFAR-10 and CelebA, achieving training accuracies of 99.9% and 99.7% on the original noise-free images, respectively. After pre-training, both the diffusion model $p^0_{X(t)}$ and the classifiers are frozen for all subsequent experiments.

We parameterize $\phi_\theta(\cdot, t) : [0, T] \to \mathbb{R}^{|\mathcal{Y}|}$, where $|\mathcal{Y}|$ denotes the number of target classes (e.g., 4 for CelebA and 10 for CIFAR-10). We use a shallow MLP with three fully connected hidden layers of width (32, 32, 32) and ReLU activations. We also use the time-dependent score-correction framework from DG (Kim et al., 2023)[5].

We follow the same training and model-selection pipeline as that for the synthetic and NPMLE tasks, except that KL divergence is replaced by $\mathrm{FID}(\widehat{p^*_X}, p^0_X) \equiv \mathrm{FID}(\widehat{p^*_{X(0)}}, p^0_{X(0)})$. Due to the computational cost of diffusion sampling, validation $L^2$ distances are estimated using 5,000 samples from $\widehat{p^*_X}$ during training, while final reported metrics are computed with 50,000 samples from $\widehat{p^*_X}$ and 50,000 samples from $p^0_X$ following Kim et al. (2024). For simplicity, we set $\varepsilon(t) \equiv \varepsilon(0) = 0$ in all image experiments. Consequently, checkpoint and hyperparameter selection are based solely on the validation $L^2$ distance $\|\widehat{p^*_Y} - p^1_Y\|_2$.

Unlike the synthetic and NPMLE tasks, we additionally report image results by a best-single-run selection scheme. Under this scheme, each run with a different random seed is treated as an independent candidate, and we directly select the checkpoint with the lowest validation $L^2$ distance, without first averaging selected checkpoints across seeds under the same hyperparameter setting. This reflects the practical deployment setting, where one ultimately uses a single corrected generator. These best-single-run results are reported in Tables 1 and 2. We also report the corresponding five-run results under the averaged selection scheme in Table 3, where means and standard deviations across runs are provided.

## E. Diffusion Models and Time-dependent Constrained Problems

### E.1. Background

**Score-Based Diffusion Models.** Let $p^0_X$ denote the data distribution we aim to model. Score-based diffusion models (Song et al., 2021) define a forward diffusion process that gradually perturbs $\boldsymbol{x}_0 \sim p^0_X$ into a noise variable $\boldsymbol{x}_T$, and generate data by simulating the corresponding reverse-time process (Anderson, 1982). Specifically, the forward process is modeled as the solution to the stochastic differential equation (SDE)

$$d\boldsymbol{x}_t = f(\boldsymbol{x}_t, t)dt + g(t)d\boldsymbol{w}_t, \tag{16}$$

where $\boldsymbol{w}_t$ is a standard Wiener process. The drift coefficient $f$ and diffusion coefficient $g$ are typically chosen so that $p_t(\boldsymbol{x}_t \mid \boldsymbol{x}_0)$ is Gaussian with known mean $\mu_t(f, g, \boldsymbol{x}_0)$ and covariance $\Sigma_t(f, g, \boldsymbol{x}_0)$. The corresponding reverse-time SDE, which defines the data generation process, is

$$d\boldsymbol{x}_t = \left[ f(\boldsymbol{x}_t, t) - g^2(t)\nabla_{\boldsymbol{x}_t} \log p_t(\boldsymbol{x}_t) \right] d\bar{t} + g(t)d\bar{\boldsymbol{w}}_t, \tag{17}$$

where $\bar{\boldsymbol{w}}_t$ is a standard Wiener process in reverse time $\bar{t} \in [0, T]$. Once the time-dependent score $\nabla_{\boldsymbol{x}_t} \log p_t(\boldsymbol{x}_t)$ is available, samples from $p^0_X$ can be generated by numerically solving the reverse-time SDE, for example using Euler–Maruyama discretization.

Denoising score matching (DSM) is a standard approach to learning $\nabla_{\boldsymbol{x}_t} \log p_t(\boldsymbol{x}_t)$ with a parameterized score model $s_\eta(\boldsymbol{x}_t, t)$:

$$\mathcal{L}_{\mathrm{DSM}}(\eta; p^0_X) = \mathbb{E}_{t\sim\mathcal{U}[0,T]} \left[ \mathbb{E}_{p^0_X(\boldsymbol{x}_0)} \left[ \mathbb{E}_{p_t(\boldsymbol{x}_t|\boldsymbol{x}_0)} \left[ \lambda(t) \|s_\eta(\boldsymbol{x}_t, t) - \nabla_{\boldsymbol{x}_t} \log p_t(\boldsymbol{x}_t \mid \boldsymbol{x}_0)\|^2_2 \right] \right] \right], \tag{18}$$

---

[4] https://github.com/openai/guided-diffusion
[5] https://github.com/alsdudrla10/DG

where $\lambda(t)$ is a time-dependent weighting function.

**Time-dependent Importance Reweighting for Fair Generation.** Kim et al. (2024) study the weakly-supervised fair generation problem (Choi et al., 2020) with a focus on diffusion models. Their setting assumes abundant samples from a biased distribution $p_X^0$ and scarce samples from a target fair distribution $p_X^1 \neq p_X^0$. A direct Importance-reWeighting (IW) approach applies a time-independent density ratio to the DSM objective:

$$\mathcal{L}_{\text{IW-DSM}}(\eta; p_X^0, w_\theta) = \mathbb{E}_{t \sim \mathcal{U}[0,T]} \left[ \mathbb{E}_{p_X^0(\boldsymbol{x}_0)} \left[ w_\theta(\boldsymbol{x}_0) \mathbb{E}_{p_t(\boldsymbol{x}_t|\boldsymbol{x}_0)} \left[ \lambda(t) \| s_\eta(\boldsymbol{x}_t, t) - \nabla_{\boldsymbol{x}_t} \log p_t(\boldsymbol{x}_t \mid \boldsymbol{x}_0) \|_2^2 \right] \right] \right], \quad (19)$$

where $w_\theta(\cdot)$ is trained in advance, for example by binary classification (Sugiyama et al., 2012), to estimate the density ratio $p_X^1(\cdot)/p_X^0(\cdot)$.

A key observation of Kim et al. (2024) is that estimating this ratio directly can be difficult when there is a *density chasm* (Rhodes et al., 2020) between $p_X^0$ and $p_X^1$. This difficulty can be alleviated along the diffusion path, since adding noise increases the overlap between the two distributions. They therefore introduce a time-dependent density ratio

$$\frac{p_t^1(\boldsymbol{x}_t)}{p_t^0(\boldsymbol{x}_t)} = \frac{\int p_X^1(\boldsymbol{x}_0) p_t(\boldsymbol{x}_t \mid \boldsymbol{x}_0) \, d\boldsymbol{x}_0}{\int p_X^0(\boldsymbol{x}_0) p_t(\boldsymbol{x}_t \mid \boldsymbol{x}_0) \, d\boldsymbol{x}_0},$$

represented by $w_\theta(\boldsymbol{x}_t, t)$. This ratio is learned by minimizing the time-dependent Binary Cross-Entropy (BCE) loss

$$\mathcal{L}_{\text{BCE}}(\theta; p_X^0, p_X^1) = \int_0^T \lambda'(t) \left[ \mathbb{E}_{p_t^0(\boldsymbol{x}_t)}[-\log d_\theta(\boldsymbol{x}_t, t)] + \mathbb{E}_{p_t^1(\boldsymbol{x}_t)}[-\log(1 - d_\theta(\boldsymbol{x}_t, t))] \right] dt, \quad (20)$$

where $\lambda'(t)$ is a weighting function, and

$$w_\theta(\boldsymbol{x}_t, t) = \frac{d_\theta(\boldsymbol{x}_t, t)}{1 - d_\theta(\boldsymbol{x}_t, t)}.$$

A central result of Kim et al. (2024, Theorem 1) is that $\mathcal{L}_{\text{IW-DSM}}$ can be rewritten, up to a normalizing constant, in the following Time-dependent Importance-reWeighting (TIW) form:

$$\begin{aligned}
&\mathcal{L}_{\text{TIW-DSM}}(\eta; p_X^0, w_\theta) \\
&= \mathbb{E}_{t \sim \mathcal{U}[0,T]} \left[ \mathbb{E}_{p_X^0(\boldsymbol{x}_0)} \left[ \mathbb{E}_{p_t(\boldsymbol{x}_t|\boldsymbol{x}_0)} \left[ \lambda(t) w_\theta(\boldsymbol{x}_t, t) \| s_\eta(\boldsymbol{x}_t, t) - \nabla_{\boldsymbol{x}_t} \log p_t(\boldsymbol{x}_t \mid \boldsymbol{x}_0) - \nabla_{\boldsymbol{x}_t} \log w_\theta(\boldsymbol{x}_t, t) \|_2^2 \right] \right] \right].
\end{aligned} \quad (21)$$

This reformulation makes the time-dependent density ratio explicit in the objective, and benefits from the improved accuracy of $w_\theta(\boldsymbol{x}_t, t)$ along the diffusion trajectory.

They further show in the ablation study that replacing $w_\theta(\boldsymbol{x}_t, t)$ in the weights of (21) by equal weights (EW) of 1 leads to comparable results:

$$\begin{aligned}
&\mathcal{L}_{\text{TIW-DSM-EW}}(\eta; p_X^0, w_\theta) \\
&= \mathbb{E}_{t \sim \mathcal{U}[0,T]} \left[ \mathbb{E}_{p_X^0(\boldsymbol{x}_0)} \left[ \mathbb{E}_{p_t(\boldsymbol{x}_t|\boldsymbol{x}_0)} \left[ \lambda(t) \| s_\eta(\boldsymbol{x}_t, t) - \nabla_{\boldsymbol{x}_t} \log p_t(\boldsymbol{x}_t \mid \boldsymbol{x}_0) - \nabla_{\boldsymbol{x}_t} \log w_\theta(\boldsymbol{x}_t, t) \|_2^2 \right] \right] \right].
\end{aligned} \quad (22)$$

As an alternative to optimizing $\mathcal{L}_{\text{TIW-DSM-EW}}(\eta; p_0, w_\theta)$ directly, this form suggests a score-corrected sampling scheme – to see this, replace $s_\eta(\boldsymbol{x}_t, t) - \nabla_{\boldsymbol{x}_t} \log w_\theta(\boldsymbol{x}_t, t)$ with $s_{\eta,\theta}'(\boldsymbol{x}_t, t)$ in $\mathcal{L}_{\text{TIW-DSM-EW}}(\eta; p_0, w_\theta)$. The objective is then identical to (19), and thus $s_{\eta,\theta}'(\boldsymbol{x}_t, t)$ converges to $\nabla_{\boldsymbol{x}_t} \log p_t^0(\boldsymbol{x}_t)$. It follows that $s_{\eta,\theta}'(\boldsymbol{x}_t, t) + \nabla_{\boldsymbol{x}_t} \log w_\theta(\boldsymbol{x}_t, t)$ converges to $\nabla_{\boldsymbol{x}_t} \log p_t^0(\boldsymbol{x}_t) + \nabla_{\boldsymbol{x}_t} \log \frac{p_t^1(\boldsymbol{x}_t)}{p_t^0(\boldsymbol{x}_t)} = \nabla_{\boldsymbol{x}_t} \log p_t^1(\boldsymbol{x}_t)$. In other words, one can train $s_\eta(\boldsymbol{x}_t, t)$ with (18) and $w_\theta(\boldsymbol{x}_t, t)$ with (20) separately; at inference time, use the corrected score $s_\eta(\boldsymbol{x}_t, t) + \nabla_{\boldsymbol{x}_t} \log w_\theta(\boldsymbol{x}_t, t)$ for sampling. Nonetheless, such an approach trades off inference-time for flexibility and re-usability.

**Time-dependent Discriminator Guidance.** There is a closely related earlier work by Kim et al. (2023). In this study, the authors aim to improve the sample quality of a given diffusion model with score network $s_\eta(\boldsymbol{x}_t, t)$ by similar score adjustment. To do this, they first learn density ratio $w_\theta(\boldsymbol{x}_t, t) \approx \frac{p_t^1(\boldsymbol{x}_t)}{p_t^0(\boldsymbol{x}_t)} = \frac{\int p_X^1(\boldsymbol{x}_0) p_t(\boldsymbol{x}_t|\boldsymbol{x}_0) \, d\boldsymbol{x}_0}{\int p_X^0(\boldsymbol{x}_0) p_t(\boldsymbol{x}_t|\boldsymbol{x}_0) \, d\boldsymbol{x}_0}$, where, in this case, $p_X^1$ is the empirical distribution of real data and $p_X^0$ is the data distribution associated with $s_\eta(\boldsymbol{x}_t, t)$, by minimizing the BCE loss (20). At inference time, samples are generated using the corrected score $s_\eta(\boldsymbol{x}_t, t) + \nabla \log w_\theta(\boldsymbol{x}_t, t)$.

### E.2. Time-dependent Constrained Problem for Diffusion Models

In this section, we study the constrained problem (1) when the reference distribution $p_X^0$ is given by a pre-trained diffusion model with parameterized score function $s_\eta(\boldsymbol{x}_t, t)$. A naïve approach, analogous to (19), is to correct the score for $t = 0$ only. However, this introduced a sudden discontinuity in the corrected scores at $t = 0$: $\nabla_{\boldsymbol{x}_0} \log p_{\boldsymbol{x}_0}(\boldsymbol{x}_0; \theta) = s_\eta(\boldsymbol{x}_0, 0) + w_\theta(\boldsymbol{x}_0)$ is adjusted while $\nabla_{\boldsymbol{x}_t} \log p_{X(t)}(\boldsymbol{x}_t) = s_\eta(\boldsymbol{x}_t, t)$ remains intact for $t > 0$. Inspired by (Kim et al., 2024; 2023), we consider an extended time-dependent constrained problem:

$$
\begin{aligned}
p_{X(t)}^* = \underset{p_{X(t)} \in \mathcal{P}_\mathcal{X}}{\arg\min} \quad & D_{\mathrm{KL}}(p_{X(t)} \| p_{X(t)}^0) \\
\text{subject to} \quad & \| p_Y^1 - p_{Y|X(t)} \circ p_{X(t)} \|_2 \le \varepsilon(t)
\end{aligned}
\tag{23}
$$

for $t \in [0, T]$, where $p_{X(t)}^0$ is characterized by the diffusion process with $s_\eta(\boldsymbol{x}_t, t)$ and a terminal noise $\boldsymbol{x}_T$. Note that our ultimate goal (1) is recovered at $t = 0$; for $t > 0$, we want to relax our problem such that $w_\theta(\boldsymbol{x}_t, t) \approx \frac{p_{X(t)}^*(\boldsymbol{x}_t)}{p_{X(t)}^0(\boldsymbol{x}_t)} = \frac{\int p_{X(0)}^*(\boldsymbol{x}_0) p_t(\boldsymbol{x}_t | \boldsymbol{x}_0)\, d\boldsymbol{x}_0}{\int p_{X(0)}^0(\boldsymbol{x}_0') p_t(\boldsymbol{x}_t | \boldsymbol{x}_0')\, d\boldsymbol{x}_0'}$ is smooth in time $t$ and easy to estimate. In addition, one can further relax the problem by a time-dependent tolerance $\varepsilon(t) \ge \varepsilon$ provided that $\varepsilon(0) = \varepsilon$.

Unlike Kim et al. (2024) where a target distribution $p_X^1$ is available as scarce samples, our target solution $p_{X(t)}^*$ is unavailable both as a known distribution and and as samples, and consequently we cannot learn $w_\theta(x_t, t)$ by optimizing a BCE objective such as (20). Instead, we train the time-dependent dual variable $\phi_\theta(\boldsymbol{x}_t, t)$ by minimizing the time-averaged negated dual objective

$$
L(\theta) = \mathbb{E}_{t \sim \mathcal{U}[0, T]}[\lambda(t) L(\theta, t)]
$$

where

$$
\begin{aligned}
L(\theta, t) = \Bigg\{ & \varepsilon(t) \left( \int_\mathcal{Y} \phi_\theta^2(y, t)\, dy \right)^{\frac{1}{2}} - \int_\mathcal{Y} \phi_\theta(y, t) p_Y^1(y)\, dy + \log \int_\mathcal{X} p_{X(t)}^0(\boldsymbol{x}_t) \cdot \exp\left( \int_\mathcal{Y} \phi_\theta(y, t) p_{Y|X(t)}(y | \boldsymbol{x}_t)\, dy \right) d\boldsymbol{x}_t \Bigg\} \\
& := L_{\mathrm{reg}}(\theta, t) + L_1(\theta, t) + L_2(\theta, t),
\end{aligned}
\tag{24}
$$

and $\lambda(t)$ is a temporal weighting function.

The training scheme remains identical to the one in Section 3, except for the additional time argument. For instance, the gradient of $L_2(\theta, t)$ is given by:

$$
\nabla_\theta L_2(\theta, t) = \int_\mathcal{X} \int_\mathcal{Y} p_{X(t)}(\boldsymbol{x}_t; \theta) p_{Y|X(t)}(y | \boldsymbol{x}_t) \nabla_\theta \phi_\theta(y, t)\, dy\, d\boldsymbol{x}_t.
\tag{25}
$$

Thus, given samples $\boldsymbol{x}_t \sim p_{X(t)}(\cdot; \theta)$ and, for continuous $Y$, samples $y \sim p_{Y|X(t)}(\cdot \mid \boldsymbol{x}_t)$, we can estimate the gradient exactly as before. For discrete $Y$, as in our image experiments, the integral over $\mathcal{Y}$ is a finite sum and no Monte Carlo sampling over $Y$ is required.

It remains to describe how to sample from $p_{X(t)}(\cdot; \theta)$. Note that $p_{X(t)}(\boldsymbol{x}_t; \theta) \propto p_{X(0)}(\boldsymbol{x}_0; \theta) p_t(\boldsymbol{x}_t | \boldsymbol{x}_0)$ implies a two-stage sampling scheme – we first sample $\boldsymbol{x}_0 \sim p_{X(0)}(\boldsymbol{x}_0; \theta)$ by invoking the diffusion sampler with the adapted score function defined below; the full procedure is given in Algorithm 3:

$$
\begin{aligned}
\nabla_{\boldsymbol{x}_t} \log p_{X(t)}(\boldsymbol{x}_t; \theta) &= \nabla_{\boldsymbol{x}_t} \log p_{X(t)}^0(\boldsymbol{x}_t) + \nabla_{\boldsymbol{x}_t} \log w_\theta(\boldsymbol{x}_t, t) \\
&= s_\eta(\boldsymbol{x}_t, t) + \nabla_{\boldsymbol{x}_t} \int_\mathcal{Y} \phi_\theta(y, t) p_{Y|X(t)}(y | \boldsymbol{x}_t)\, dy \\
&= s_\eta(\boldsymbol{x}_t, t) + \int_\mathcal{Y} \phi_\theta(y, t) \nabla_{\boldsymbol{x}_t} p_{Y|X(t)}(y | \boldsymbol{x}_t)\, dy \\
&= s_\eta(\boldsymbol{x}_t, t) + \int_\mathcal{Y} \phi_\theta(y, t) \cdot \nabla_{\boldsymbol{x}_t} \log p_{Y|X(t)}(y | \boldsymbol{x}_t) \cdot p_{Y|X(t)}(y | \boldsymbol{x}_t)\, dy,
\end{aligned}
\tag{26}
$$
$$
\tag{27}
$$

followed by adding time-dependent noise according to $p_t(\boldsymbol{x}_t | \boldsymbol{x}_0)$ defined by the forward diffusion process.

At inference time, we sample data from $\boldsymbol{x}_0 \sim p_{X(0)}(\boldsymbol{x}_0; \widehat{\theta}^*)$ with the corrected score defined above without adding noise, where $\widehat{\theta}^* = \operatorname{argmax}_\theta L(\theta)$.

When computing the corrected score $\nabla_{\boldsymbol{x}_t} \log p_{X(t)}(\boldsymbol{x}_t; \theta)$, we use (27) for continuous $Y$, since it admits a Monte Carlo approximation by sampling $y \sim p_{Y|X(t)}(\cdot|\boldsymbol{x}_t)$. For discrete $Y$ (e.g., our image tasks), however, direct computation via (26) is preferable, since the integral reduces to an exact and efficient dot product between two finite-dimensional real vectors. In particular, when (26) is employed, the diffusion sampler still requires $\mathcal{O}(T)$ score evaluations for a $T$-step trajectory, but each step additionally evaluates the time-dependent classifier $p_{Y|X(t)}(y|\boldsymbol{x}_t)$ and its gradient with respect to $\boldsymbol{x}_t$. Empirically, this roughly doubles the wall-clock sampling time for our image tasks; see Appendix C.2 for a more detailed analysis.

## F. Algorithm Details

We describe the implementation of **DUALSCOREGENX** for both SGLD and diffusion models.

---

**Algorithm 2 DUALSCOREGENX** for SGLD at the $k$th iteration.

---

**function DUALSCOREGENX**$(\theta^{(k)}, x^{(k)}, L = 1)$

    **Initialize:** $x_0 = x^{(k)}$    {Intialize persistant chain}

    **for** $i = 1, \ldots, L + N$ **do**

        Sample $\{y_{4,j}\}_{j=1}^{J} \sim p_{Y|X}(\cdot \mid x_{i-1})$ and $z_i \sim \mathcal{N}(0, I)$

        $x_i \leftarrow x_{i-1} + \alpha \left( \nabla_x \log p_X^0(x_{i-1}) + \frac{1}{J} \sum_{j=1}^{J} \phi_{\theta^{(k)}}(y_{4,j}) \cdot \nabla_x \log p_{Y|X}(y_{4,j}|x_{i-1}) \right) + \sqrt{2\alpha} z_i$

    **end for**

    **return** $\{x_i\}_{i=L+1}^{L+N}$

**end function**

---

---

**Algorithm 3 DUALSCOREGENX** for score-based diffusion models at the $k$th iteration.

---

**function DUALSCOREGENX**$\left(\theta^{(k)}, N, \{t_i\}_{i=1}^{N}\right)$

    Sample diffusion times $\{t_i\}_{i=1}^{N} \sim \text{Unif}(0, T)$

    **for** $i = 1, \ldots, N$ **do**

        **Initialize:** $x_i^T \sim \mathcal{N}(0, I)$

        **for** $\tau = T, T - \Delta t, \ldots, t_i + \Delta t$ **do**

            Compute corrected score: $s_{\text{corr}}(x_i^\tau, \tau) \leftarrow s_\eta(x_i^\tau, \tau) + \sum_{y \in \mathcal{Y}} \phi_{\theta^{(k)}}(y, \tau) \nabla_{x_i^\tau} p_{Y|X(\tau)}(y \mid x_i^\tau)$

            Take one reverse diffusion step: $x_i^{\tau - \Delta t} \leftarrow \textbf{ReverseDiffusionStep}\,(x_i^\tau, s_{\text{corr}}(x_i^\tau, \tau))$

        **end for**

        Set $x_i \leftarrow x_i^{t_i}$

    **end for**

    **return** $\{x_i\}_{i=1}^{N}$

**end function**

---

