# OpenReview forum: "Score Correction for Generative Models with Probabilistic Constraints"
_ICML.cc/2026/Conference — ICML 2026 regular_

### Official Review · Reviewer_QLzt · 2026-03-08

**Soundness:** 4
**Presentation:** 3
**Significance:** 4
**Originality:** 4
**Overall Recommendation:** 5
**Confidence:** 3

**Summary:**

This paper formalizes the probabilistic constraints on generative models through a KL-divergence minimization problem with $L_2$ constraints on the induced marginal distributions. The authors propose to solve it by optimizing the dual of the constrained problem, with a neural network parameterizing the infinite-dimensional dual variable. The practical solution involves a combination of Stochastic Gradient Langevin Dynamics (SGLD) with Stochastic Gradient Descent (SGD) to fit the dual parameters and later simulate samples from the adjusted distribution. The performance of this approach is demonstrated on both synthetic and real-world tasks.

**Compliance With Llm Reviewing Policy:**

Affirmed.

**Final Justification:**

As mentioned in the rebuttal acknowledgment, I maintain the positive score as it properly reflects my view on the paper's value.

**Key Questions For Authors:**

Q1. Please refer to W1.

Q2. Please refer to W2.

Q3. Please refer to W3.

**Limitations:**

yes

**Strengths And Weaknesses:**

S1 (Soundness): The proposed approach of handling infinite-dimensional constraints through dual optimization is formally grounded and principled.

S2 (Presentation): The paper is well-structured, with rigorous approach to the definition of each notion of the problem.

S3 (Significance): The considered problem is important due to frequently unmet constraints of generative models regarding fairness or domain-specific adaptations. Moreover, the proposed solution does not require model retraining, relying on the use of an additional light-weight neural network for `correction`.

S4 (Originality): The paper combines well-established tools (SGLD, dual optimization) in a creative manner, offering a novel and practical solution.

W1: Real-time applicability of the proposed approach may be limited in high-dimensional spaces (e.g., high-res images) due to the persistent Markov chain slowing down the standard sampling process (e.g., of a diffusion model). I would appreciate a formal comment on the resulting complexity of the adjusted sampling procedure.

W2: How does the bias introduced by the square-root term affect the final dual solution? I agree that weighted averaging is an effective approach, but would appreciate a further formal analysis.

W3: The authors state that, under the infeasible problem, the dual diverges, mentioning their method as a good diagnostic tool. Are there any guarantees on the method achieving some "best-effort" solution when the constraint is impossible to fulfill?

---

> ### Author Rebuttal · Authors · 2026-03-31
>
> Thank you for your positive and constructive comments. To answer your questions:
>
> **Effect of the persistent Markov chain:**
>
> Please see our first response to Reviewer **6H56**
>
> **W1: Complexity of sampling in data dimension:**
>
> The adjusted sampling procedure has the same order of complexity with respect to data dimensionality as the unadjusted sampler. For SGLD, drawing samples with a length-$T$ trajectory has cost $O\left(TC_0\right)$ for the unadjusted sampler and $O\left(T(C_0 + C_{\mathrm{corr}})\right)$ for the adjusted sampler, where $C_0$ is the cost of evaluating the base score $\nabla_x \log p^0_X(x)$ and $C_{\mathrm{corr}}$ is the cost of the correction term in Eq. 8. Both $C_0$ and $C_{\mathrm{corr}}$  are model-dependent quantities. We consider the Gaussian and diffusion settings separately. Below, $d_X := \dim(X)$  and  $d_Y := \dim(Y)$.
>
> - **Gaussian experiments:** $C_0 = O(d_X^2)$ for a closed-form Gaussian score with dense covariance, and $O(d_X)$ under independence / diagonal covariance. For $C_{\mathrm{corr}}$, if the expectation in Eq. 8 is estimated with $M$ Monte Carlo samples from $p_{Y\mid X}$, then $C_{\mathrm{corr}} = M C_{Y|X}$ , where $C_{Y|X}$ is the cost of drawing one sample from $p_{Y\mid X}$ and computing $\nabla_x \log p_{Y\mid X}(y\mid x)$. For example, for a linear-Gaussian conditional $p_{Y\mid X}(\cdot\mid x)=\mathcal N(Ax+b,\Gamma)$, we have $\nabla_x \log p_{Y\mid X}(y\mid x)=A^\top\Gamma^{-1}(y-Ax-b)$. After precomputing $\Gamma^{-1}$, one such evaluation costs $O(d_X d_Y)$ for dense $A$, and just $O(d_X)$ for scalar Y. Thus, the overall time complexity of the adjusted sampler is $O\left(M \cdot TC_0\right)$
> , the same order in $d_X$ as the unadjusted sampler.
> - **Diffusion models:** Here the same decomposition applies, but the sampling procedure is the reverse diffusion trajectory rather than SGLD. To draw one sample, a $T$-step reverse trajectory will cost $O\left(TC_0\right)$ for the unadjusted sampler, and $O\left(T(C_0 + C_{\mathrm{corr}})\right)$ for the adjusted one, since the correction score is evaluated once at each reverse-time step.
>
>     In our image experiments, $C_0$ is the cost of one EDM score-network evaluation on an image tensor of dim $d_X$, while $C_{\mathrm{corr}}$ is the cost of one evaluation of the score correction per Eq. 28. In our implementation, $C_\text{corr}$ is dominated by one forward pass through $p_{Y\mid X(t)}(y\mid x_t)$ and one backward pass to obtain the gradient w.r.t. $x_t$. We implement $p_{Y\mid X(t)}(y\mid x_t)$ as a smaller two-layer U-Net encoder than the EDM U-Net (Appendix C.1.2), so both have the same $d_X$ complexity for fixed $d_Y$. Further, the dependence on $d_Y$ is negligible in our image tasks because $|Y|$ is small (4 for CelebA and 10 for CIFAR-10). Therefore, the overall time complexity is $O\left(T(C_0 + C_{\mathrm{corr}})\right)=O\left(2\cdot TC_0\right)$ in our implementation (which is empirically supported by the observation reported at the end of Appendix D.2). Again, the adjusted sampler has the same order in $d_X$ as the unadjusted sampler.
>
> We will clearly include this in the supplementary material of the revision.
>
> **W2: Bias of the square-root term:**
>
> In our experiments, since we use light-weight networks for $\phi_\theta$ , our approach did not result in too much variance in the estimated norm, and thus not too much bias. It is worth noting that by a simple application of Jensen's inequality, the bias induced by the square-root term is conservative, causing the marginal constraint to be enforced more strongly than required by the problem specification. This is usually acceptable, as nonzero $\varepsilon$ is typically introduced to relax the problem when feasibility is unknown. (in fact, when $\varepsilon=0$, the norm no longer appears in the dual and the bias disappears). This is consistent with our empirical results (e.g., Fig.3): across the relaxed-constraint experiments, the returned solutions are typically very close to the prescribed constraint boundary, and in some cases slightly below it. We will discuss this and include a proof in the appendix of the revised version.
>
> **W3: Guarantees on achieving "best-effort" solution**
>
> We do not currently claim a formal “best-effort” guarantee in the infeasible regime, this was just an informal conjecture based on empirical results. For example, in the NPMLE experiment with $\varepsilon=0$, the achieved marginal discrepancy is typically lower than in the runs with fixed nonzero $\varepsilon$, even if exactly satisfying the constraint is infeasible. We will revise the paper to clarify this.
>
> Our current approach does support the use of divergence of the dual optimization as a diagnostic of infeasibility, rather than a guarantee on the optimality of the terminated iterate. This can be used to perform e.g. a binary search over $\varepsilon$, to find an approximation to the smallest feasible value.

---

> > ### Author Rebuttal · Reviewer_QLzt · 2026-04-02
> >
> > I thank the authors for their detailed response; all of my concerns have been addressed properly. I believe that my current score correctly reflects the soundness of the paper; hence, I will maintain it.

---

### Official Review · Reviewer_6H56 · 2026-03-11

**Soundness:** 2
**Presentation:** 2
**Significance:** 3
**Originality:** 2
**Overall Recommendation:** 4
**Confidence:** 3

**Summary:**

The paper addresses the challenge of modifying black-box generative models to satisfy probabilistic marginal constraints derived from domain knowledge. The authors formally analyze this problem, noting that exact matching of a constraint density is often intrinsically infeasible for general stochastic transformations. Consequently, they relax the objective into an inequality-constrained optimization problem bounded by a tolerance parameter $\epsilon$. They acknowledge that determining the optimal feasibility threshold $\epsilon^\star$​ is challenging and do not explicitly solve for it, leaving it as a relaxed bound.

Rather than retraining a model from scratch, the approach assumes access to a base reference density. Because the direct constrained optimization over the infinite-dimensional probability simplex is intractable, the authors cleverly optimize its dual formulation, parameterizing the infinite-dimensional dual variable with a neural network. This yields a loss function amenable to stochastic gradient descent via Monte Carlo estimation. To overcome the "density chasm" issues that severely degrade standard importance sampling approaches, the authors employ Stochastic Gradient Langevin Dynamics (SGLD) to sample directly from the adapted density during training. This requires access to the score functions of both the reference and transition densities. To maintain computational efficiency across these SGLD steps, they utilize a persistent Markov chain throughout the training process.

Finally, the proposed method is validated empirically on synthetic tasks and two real-world applications: a regularized nonparametric maximum likelihood estimation (NPMLE) problem, and enforcing class-level fairness constraints in image diffusion models (e.g., balancing attributes in CelebA and CIFAR-10).

**Compliance With Llm Reviewing Policy:**

Affirmed.

**Final Justification:**

The authors successfully addressed many of my concerns, so I have decided to increase my score to a '4: Weak Accept,' which I consider a recommendation for publication. However, my concerns regarding the estimated impact (significance) and limited empirical scope remain valid, hence the 'weak' designation. I want to thank the authors for their constructive engagement in this process.

**Key Questions For Authors:**

1. Variance of the L_2​ gradient estimator: In Algorithm 1, the gradient estimator for L_2​ relies on drawing a single sample $y_i \sim ​p_{Y∣X​}(\cdot∣x_i​)$ for each $x_i$​ in the SGLD batch. While unbiased, wouldn't this estimator suffer from high variance if the transition density $p_{Y∣X​}$ is highly stochastic or multimodal, especially given the small batch sizes used in the experiments? Have you experimented with taking multiple y samples per xi​ to stabilize training in such scenarios?
2. Initialization details: The manuscript states that the model is initialized such that $\nabla_x \log p^1_X(x; \theta^{(0)}) \approx \nabla_x \log p^0_X(x)$ Based on Equation 8, this implies the neural network $\phi_\theta(y)$ must be initialized to output near-zero values. Could the authors clarify the exact weight initialization scheme used for $\phi_\theta(y)$ to achieve this, and add this detail to the reproducibility section in the appendix?

**Limitations:**

yes

**Strengths And Weaknesses:**

**Soundness**

*Strengths:*

- The core theoretical transition from an intractable infinite-dimensional primal problem to a tractable dual formulation is mathematically sound and elegantly executed.
- The authors correctly identify the "density chasm" problem that plagues standard importance sampling techniques and effectively motivate the use of Stochastic Gradient Langevin Dynamics (SGLD) to overcome it. The derivation of the unbiased gradient estimators for the SGLD approach is solid.

*Weaknesses:*

- Unaddressed biases in persistent chain sampling: To make SGLD computationally feasible, the authors maintain a persistent Markov chain during training. They initialize the current step's chain with the final state of the previous iteration's chain, assuming the distributions before and after an SGD step are similar. While this reduces burn-in time, it effectively implements Persistent Contrastive Divergence (PCD) [1] (see the 'Presentation' section for further discussion) without addressing the associated theoretical caveats.
- Lack of empirical validation for the sampling dynamics: Because the samples are Markovian, they carry a "memory" of previous model states. If the SGD learning rate is too high or the Langevin mixing is too slow, the chain will lag behind the model, leading to stale, biased gradient estimates. The submission lacks formal discussion and empirical validation regarding this dynamic. A sensitivity analysis showing how the number of SGLD steps per iteration interacts with the SGD learning rate is necessary to prove that the model does not suffer from historical bias or the "stale sample" problem addressed in later PCD literature [2].

**Presentation**

*Strengths:*

- The paper is generally well-written and logically structured. The problem setup is formalized clearly, and the limitations of deterministic constraint baselines when applied to stochastic transformations are illustrated well.
- Algorithm 1 provides a concise and helpful overview of the training and simulation loop.

*Weaknesses:*

- Missing attribution for foundational techniques: The presentation of the persistent chain methodology is currently stated simply as an "in practice" implementation detail. The authors fail to connect this to the established literature on persistent chains. The manuscript must cite and discuss the foundational works that introduced and analyzed these dynamics, specifically:

      [1] Tieleman, Tijmen. "Training restricted Boltzmann machines using approximations to the likelihood gradient." Proceedings of the 25th international conference on Machine learning. 2008.

      [2] Tieleman, Tijmen, and Geoffrey Hinton. "Using fast weights to improve persistent contrastive divergence." Proceedings of the 26th annual international conference on machine learning. 2009.

- By omitting this context, the presentation glosses over the critical hyperparameters required to keep the evolving equilibrium stable, making the methodology section feel incomplete.
- Minor typographical and presentational errors: The manuscript would benefit from a proofreading pass to address several minor issues:
   * Line 077 (left column): Phrasing "such as a diffusion model or a transformer"  is an "apples to oranges" comparison, as one is a generative framework and the other is a specific neural architecture.
   * Line 092 (right column): The colloquial transition "Now," at the beginning of the sentence ("Now, constraints are typically understood...")  should be removed.
   * Line 131 (right column): Typographical error: "practitioners must by careful"  should be "must be careful".
   * Line 203 (right column): There is a contradiction regarding the optimizer; the text explicitly defines the standard stochastic gradient descent (SGD) update rule but then immediately states "We use Adam... in our implementation.".
   * Lines 1013-1014 (Appendix D.1): An unresolved author note/TODO remains in the text: "[SW: Fix this]".
   * Algorithm 1 & 2: In Algorithm 1 (Lines 237-238) and Algorithm 2 (Line 1526), the state update for the persistent chain $x^{(t)}$ is currently denoted as updating to $x_N​$. This should be corrected to explicitly show it updating with the set obtained from the SGLD step earlier in the loop.

**Significance**

*Strengths:*

- The paper addresses a mathematically interesting and practically relevant niche: steering generative models to respect specific probabilistic constraints derived from domain knowledge.
- While perhaps not a universally encountered bottleneck, this framework has valid and important applications in specialized domains. The authors demonstrate its utility in scenarios like enforcing class-level fairness in image generation and addressing regularized nonparametric maximum likelihood estimation (NPMLE). Furthermore, as the authors note, it has potential implications for privacy-preserving data synthesis.
- The paper successfully highlights the inadequacy of applying deterministic constraint solutions (like conditional sampling or simple importance reweighting) to stochastic transformation problems, identifying a genuine gap in the current literature.

*Weaknesses:*
- The scope of the problem is somewhat specialized. The requirement of having both a known transition density and a target marginal constraint density means this method will likely serve a relatively narrow subset of practitioners working on highly specific alignment or regularization tasks.

**Originality:**

*Strengths:*

- The approach to solving the constrained optimization problem is clever. Rather than fighting the intractable infinite-dimensional primal problem, leveraging the dual formulation and parameterizing the dual variable with a neural network is a creative and elegant bridge between classical duality and modern deep learning.
- Recognizing the "density chasm" issue inherent to importance sampling and pivoting to an SGLD-based sampling approach demonstrates a strong understanding of the practical limitations of existing sampling techniques.

*Weaknesses:*
- Undermined by omitted literature: The originality of the specific training and sampling mechanism is difficult to fully credit due to the omission of foundational literature. As noted in the Soundness and Presentation sections, the authors' use of a persistent Markov chain initialized from the previous iteration's state is essentially Persistent Contrastive Divergence (PCD). By introducing this merely as an "in practice" detail without citing the original PCD works, the authors obscure the lineage of their method. This makes it challenging to disentangle what is truly a novel algorithmic contribution versus an uncredited application of an existing technique.

---

> ### Author Rebuttal · Authors · 2026-03-31
>
> Thank you very much for your thorough and constructive review of our submission.
>
> **Soundness, Presentation** and **Originality:** The main weakness you raised under all these three criteria concerned the connection between our training scheme and Persistent Contrastive Divergence (PCD), and its implications on the SGD learning rate. You are correct that we should have cited this literature, and we thank you for raising this. Indeed, while PCD is traditionally formulated with Gibbs updates and we use SGLD, both are instances of persistent MCMC-based gradient methods. More broadly, such approaches fit within the framework of two-timescale stochastic approximation [1], where a “fast” Markov chain tracks the evolving target distribution defined by a “slow” parameter update (used e.g. in actor-critic learning). We will revise the manuscript to include this discussion and relevant citations.
>
> As noted by you and Reviewer **QLzt**, the important implication of this connection is the fact that instability can arise if the SGD step size is too large relative to the mixing rate of the SGLD sampler, leading to biased (“stale”) gradient estimates. As you requested, we have included an additional set of results, for the synthetic density-chasm experiment in Section 4.1. The results indeed show an interaction between Langevin mixing and the learning rate: with a large learning rate $10^{-3}$, training is more sensitive to the number of SGLD steps, whereas with a smaller learning rate $10^{-4}$, the method remains stable across a broad range of SGLD steps and achieves similarly low final constraint error. Please see the anonymous figure here: https://figshare.com/s/313c61065cb88e4c8846.
>
> Our original experiments avoided this issue through standard hyperparameter tuning of the SGD learning rate. Making the connection to the PCD and stochastic approximation literature explicit is valuable, as it provides practitioners with clearer guidance and diagnostic tools for identifying and addressing such issues. We will incorporate this discussion, along with pointers to recent theoretical work [2].
>
> Finally, we mention that this issue does not arise in our diffusion model experiments, where the learned score is used to convert noise into *exact* samples, rather than to target a distribution using SGLD.
>
> We hope our response has alleviated some of your concerns about the soundness, presentation and originality of our work. Thank you also for pointing out the typographical errors which we have fixed.
>
> **Significance:** Here, you listed as a weakness that our problem is somewhat specialized, requiring a known transition density and a target marginal constraint density. We do not dispute the latter fact, though we hope our experiments show this still includes a useful range of problems. We also believe our theoretical work about stochastic-vs-deterministic transitions is relevant to current practice.
>
> To answer your final two questions:
>
> **Variance in** $L_2$: For a fixed compute budget, producing more samples $y_i \sim p_{Y|X}(y_i|x_i)$
> necessarily reduces the number of $x_i$  samples. We found this inadvisable in our problem setting, where $X$ is the higher-dimensional and more complex variable, and accurate exploration of its distribution is more critical for stable training. We agree that when $p_{Y|X}$ is very noisy, multiple $y_i$ samples per $x_i$ can reduce gradient variance. However, since $p_{Y|X}$ is a specified input, the practitioner can directly assess its stochasticity and increase the number of samples $M$ accordingly. We will revise the paper to clarify that $M=1$ was chosen as an efficient default in our experiments rather than a universal choice.
>
> As you requested, we repeated the synthetic setting in Section 4.1 with a substantially noisier conditional distribution (variance of $9^2$ instead of $1^2$), while keeping the same $p_X^0$ and $p_X^\ast$. We varied the number of conditional samples per $x_i$ as $1$, $100$, and $10000$. The results show that using more conditional samples slightly reduces variability in the early phase of training and mitigates occasional spikes, but has little effect on the final $KL(\hat p_X^\ast \Vert p_X^0)$ and $\Vert\hat p_Y^\ast - p_Y^1\Vert_2$ values. Please see the anonymous results here: https://figshare.com/s/0b7baaaa303d8ac3667c.
>
> **Initialization so that score-network’s output is close to 0**: In our implementation, $\phi_\theta(y)$ is a standard MLP imported from TorchRL’s MLP module. After the default initialization, we scaled the initialized weights and biases by 0.1 to keeps the initial outputs close to zero.
> We will add this initialization detail to the reproducibility section (Appendix C) in the revised manuscript.
>
> [1] Borkar, V. "Stochastic approximation with two time scales." *Systems & Control Letters (1997)*
>
> [2] Oliva, V., Felix, P., Akyildiz, O., and Duncan, A.. "Uniform-in-time convergence bounds for
> Persistent Contrastive Divergence Algorithms." (2025)

---

> > ### Author Rebuttal · Reviewer_6H56 · 2026-04-02
> >
> > Thank you for the reply. I am glad the authors recognized that PCD is highly relevant to their work and will include the appropriate citations and discussion in the revised version. I also expect that the additional ablations provided in the rebuttal will find their way into the revised manuscript. In fact, I strongly encourage an even broader ablation study across multiple datasets and distributions.
> >
> > Additional questions/comments:
> > 1. "Finally, we mention that this issue does not arise in our diffusion model experiments, where the learned score is used to convert noise into exact samples, rather than to target a distribution using SGLD." — Could you please point me to where this is explicitly stated in the manuscript?
> > 2. Repeated information (Lines 1087-1090): While trying to find the answer to the question above, I noticed two consecutive sentences that convey the exact same information.
> > 3. Impact of $M$ in $L_2$​: I would expect to see the impact of $M$ evaluated in a scenario where $p_{Y∣Z}$​ is multi-modal. Adding such an experiment to the final paper would make the submission significantly stronger.

---

> > > ### Author Response · Authors · 2026-04-05
> > >
> > > Thank you for following up on this. Yes, we will include both the discussion of PCD, as well as the ablation study in the revised manuscript. We will apply the latter more broadly by also including (for other experiments) results where we disable hyperparameter tuning for the SGD step size and instead fix them across a range of values.
> > >
> > > Below are our answers to your additional questions:
> > >
> > > ***1. "Finally, we mention that this issue does not arise in our diffusion model experiments, where the learned score is used to convert noise into exact samples, rather than to target a distribution using SGLD."** **— Could you please point me to where this is explicitly stated in the manuscript?***
> > >
> > > On lines 375–377 of the manuscript, we write *"the optimization scheme described in Section 3.1 remains mostly unchanged, with the diffusion sampler having the adapted score function $s_\eta(x_t,t)+\nabla_{x_t}\log w_\theta(x_t,t)$"*. We agree that this does not explicitly state that the samples are exact or independent, rather, this is implied by the context:  diffusion models use the score function to convert noise into independent samples. Highlighting this will be especially important when we include an extended discussion about potential issues from PCD.
> > >
> > >  We will make the following changes:
> > >
> > > 1. As per our earlier discussion, add the discussion about PCD in Section 3.2 (Related work).
> > > 2. Clarify that this issue does not arise in the diffusion model experiment, where the raw and corrected score functions produce *independent* X’s rather than an SGLD Markov chain of X’s.
> > > 3. Revise Algorithm 1 and the surrounding text to frame the sampling step more generally as score-based simulation, encompassing both SGLD (MCMC) and reverse diffusion sampling (independent samples).
> > > 4. Include the ablation study as well as effect of SGD learning rate in other experiments
> > >
> > > ***2. “Repeated information (Lines 1087-1090)”***
> > >
> > > Thank you for catching this. In the time since the submission deadline, we have carefully gone over both the main manuscript and the supplementary material and corrected this as well as a number of other minor typographical issues.
> > >
> > > ***3. Impact of  M in L_2 : I would expect to see the impact of M evaluated in a scenario where $p_{Y|X}$  is multi-modal. Adding such an experiment to the final paper would make the submission significantly stronger.***
> > >
> > > Thank you for the suggestion. We ran experiments examining the impact of $M$ on performance for a multimodal $p_{Y|X}$ set to a balanced mixture of 2 Gaussians with means $2X+1-m$ and $2X+1+m$, and variance $1$, under the same setting as in the previous large-variance experiments. The results are available anonymously here: https://figshare.com/s/38d92e2d55d5e1971634. Even for a large mode separation ($m=5$), we do not see any advantage to increasing the number of samples $M$ from $p_{Y|X}$.
> > >
> > > We believe this is largely due to the fact that we can draw independent samples from $p_{Y|X},$ so that multimodality is not inherently a problem (unlike MCMC); rather it just manifests as increased variance of  $p_{Y|X}$ (which we considered in our results in our original response). As we suggested earlier,  increasing the number of samples $M$ reduces the conditional variance of the gradient estimate given $X$, but the dominant source of variance is the high-dimensional $X$. We agree though that this point is worth clarifying explicitly, and we will include these results in the main paper, systematically varying both the mode separation $m$ and the number of $Y$ samples $M$.
> > >
> > > Thank you again for your questions and comments which have helped improve our manuscript. We hope these changes address your concerns.

---

### Official Review · Reviewer_pbzS · 2026-03-13

**Soundness:** 3
**Presentation:** 3
**Significance:** 3
**Originality:** 3
**Overall Recommendation:** 4
**Confidence:** 3

**Summary:**

This paper proposes a way to modify a pretrained generative model so that certain distributional constraints are satisfied. The formulation is interesting, and the treatment of stochastic transformations makes the framework broader than some more standard approaches.

**Compliance With Llm Reviewing Policy:**

Affirmed.

**Final Justification:**

I appreciate the authors’ response and the clarifications provided in the rebuttal. Let me maintain my score.

**Key Questions For Authors:**

1. Could the authors clarify what they view as the main theoretical contribution of the paper beyond the proposed formulation?
2. Could the authors provide more details on the fairness experiments, especially the setup and evaluation protocol?
3. Could this perspective also be useful for solving high-dimensional optimal transport problems? If so, what is the advantage over more direct OT approaches?

**Limitations:**

Yes

**Strengths And Weaknesses:**

A main strength of the paper is that it gives a nice interpretation of constrained generative modeling through an optimization problem in an infinite-dimensional space. This viewpoint is interesting, and the resulting method seems reasonably general. The experiments also suggest that the approach can work in practice across several settings. My main weakness is that I was not fully convinced about what the core theoretical contribution is beyond the formulation itself. The framework is elegant, but it would help if the paper more clearly highlighted the main new theoretical insight and its significance.

---

> ### Author Rebuttal · Authors · 2026-03-30
>
> Thank you for your positive words about our manuscript. Below, we address the three questions you raised:
>
> *1. The main theoretical contribution of the paper*
>
> We agree that a key contribution is the problem formulation itself: incorporating domain knowledge into black-box generative models via stochastic marginal constraints. This encompasses a broad class of practically important problems (regularized NPMLE, fairness-constrained image generation, privacy, etc.) that prior work has not addressed in this generality.
>
> Other theoretical contributions include:
> - Presenting a dual formulation of our infinite-dimensional constrained-optimization primal problem, and exploiting it to develop a more standard neural network optimization scheme. Specifically, we devise a score-correction scheme that does not require retraining the original model, and moreover, lets us fit optimization and post-training simulation into a single framework.
> - Mitigating some of the effects of the density-chasm problem (a regime where importance-based schemes collapse) through our score-based simulation scheme.
> - Showing and formalizing that simple and commonly-used solutions to this problem such as naive importance reweighting and conditional sampling fail for stochastic constraints (which are qualitatively different from deterministic constraints). Specifically, Corollary 3.3 and Proposition 3.4 show that when $p_{Y|X}$ is non-degenerate, these methods fail to satisfy the marginal constraint.
>
> We will update our manuscript to reflect this early on, as is standard in ML papers.
>
> *2. More details on the fairness experiments, especially the setup and evaluation protocol*
>
> Our fairness experiments follow standard fair-generation settings from prior work on unbiased image generation (e.g. Appendix D.1 of [1]). For CIFAR-10, the LT/5% dataset is constructed by: (i) subsampling the original balanced training set so that the class labels follow an imbalanced long-tail distribution, yielding $\mathcal{D_\text{bias}}$ ;  (ii) subsampling the original balanced training set uniformly to create a small unbiased subset, denoted *$\mathcal{D}_{\text{ref}}$*, such that $\frac{|\mathcal{D_\text{ref}}|}{|\mathcal{D_\text{bias}}|}=5\\%$; and (iii) combining the two subsets to form $\mathcal{D_{\text{train}}} = \mathcal{D_{\text{bias}}} \cup \mathcal{D_{\text{ref}}}$. For CelebA, the original training set is used, which is naturally biased in (gender, hair color).
>
> Given a pretrained diffusion model for the original image distribution $p^0_X$, a pretrained conditional classifier $p(Y\mid X_t)$, and a target marginal distribution $p_Y^1$ (uniform over the attribute $Y$; e.g., the 10 classes in CIFAR-10), we learn a lightweight neural correction so that the reverse-time scores are adjusted according to Eq. (28) during sampling.
>
> For evaluation, we report two complementary metrics. Following [1], we generate 50,000 samples from the corrected model $\hat{p}^\ast_X$ and 50,000 samples from the original pretrained model $p_X^0$, and then compute: (i) the FID between these two sample sets, which serves as a practical proxy for $KL(\hat p_X^\ast \Vert p_X^0)$ in the primal objective; and (ii) $\Vert\hat p_Y^* - p_Y^1\Vert_2$, where $\hat p_Y^*$ is estimated from the generated samples using the pretrained conditional classifier. The second metric is our primary fairness metric, since it directly measures how closely the generated marginal over the attributes matches the desired target distribution.
>
> While these details are in Section 4.3 and Appendix C, we agree it is worth consolidating the above summary in the main document.
>
> *3. Could this perspective be useful for high-dim optimal transport problems?*
>
> Our paper already notes links to entropy-regularized OT and Schrödinger bridges, and indeed, there are meaningful connections (e.g. [2], [3]). However, classical OT is not a direct special case of our current algorithm, since in OT the coupling / transport plan $p_{Y|X}$ is the optimization object and there are two marginal constraints, whereas in our paper $p_{Y\mid X}$ is fixed and only one marginal is constrained. Viewed under our framework, the marginal constraints in OT are on deterministic projections of $(X,Y)$, which loses some of the interesting structure in our setup. We could of course consider a generalized OT problem with multiple stochastic constraints, and use our ideas (which work well in high-dim settings) e.g. as part of an iterative proportional fitting scheme. Extending our methodology to multiple constraints while maintaining its simplicity is a direction for future work.
>
> [1] Kim, Y., Na, B., Park, M., Jang, J., Kim, D., Kang, W., and Moon, I.-C. Training unbiased diffusion models from biased dataset. ICLR, 2024.
>
> [2] Korotin, A., Selikhanovych, A., Burnaev, E.. Neural Optimal Transport, ICLR 2023
>
> [3] Daniels, M., Maunu, T., and Hand, P. Score-based generative neural networks for
> large-scale optimal transport. NeurIPS, 2021

---

> > ### Author Rebuttal · Reviewer_pbzS · 2026-04-04
> >
> > I appreciate the authors’ response and the clarifications provided in the rebuttal. Based on this, I am happy to maintain my score.

---

### Decision · Program_Chairs · 2026-04-30

**Decision:**

Accept (regular)

**Comment:**

This paper tackles an important and practically relevant problem: how to adapt pretrained generative models to satisfy probabilistic constraints without retraining the base model. Reviewers generally viewed the core contribution positively, highlighting the principled dual formulation, the treatment of stochastic constraints, and the lightweight score-correction mechanism as technically solid and potentially useful for applications such as fairness-constrained generation and regularized NPMLE. The rebuttal was effective in addressing many concerns, especially around the connection to persistent-chain methods and the need for stronger implementation detail, while some weaknesses remain, most notably limited empirical breadth, some presentation issues, and a somewhat specialized scope. The overall consensus is that the paper offers a meaningful contribution that is likely to be built on by others, and is worthy of acceptance